# POGEMA: A Benchmark Platform for Cooperative Multi-Agent Navigation

**Alexey Skrynnik**
AIRI
Moscow, Russia

**Anton Andreychuk**
AIRI
Moscow, Russia

**Anatolii Borzilov**
MIPT
Moscow, Russia

**Alexander Chernyavskiy**
MIPT
Moscow, Russia

**Konstantin Yakovlev**
FRC CSC RAS, AIRI
Moscow, Russia

**Aleksandr Panov**
AIRI, MIPT
Moscow, Russia

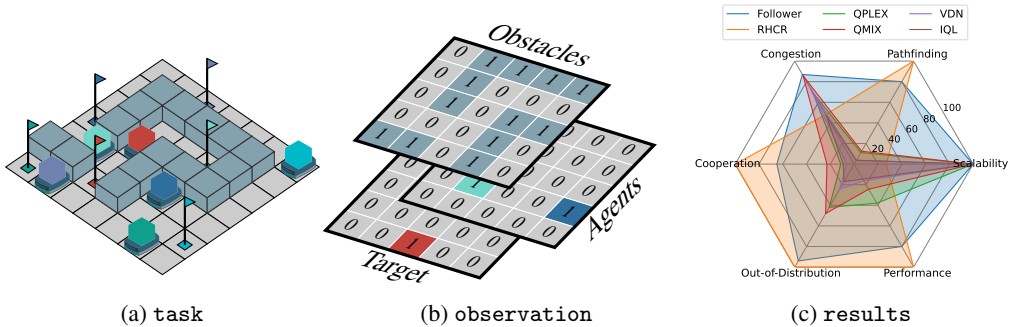

| (a) task | (b) observation | (c) results |
|---|---|---|

Figure 1: (a) Example of the multi-robot navigation problem considered in POGEMA: each robot must reach its goal, denoted by a flag of the same color. (b) Observation tensor of the red agent. (c) Results of the evaluation of several MARL, hybrid, and search-based solvers on the proposed POGEMA benchmark.

## Abstract

Multi-agent reinforcement learning (MARL) has recently excelled in solving challenging cooperative and competitive multi-agent problems in various environments with, mostly, few agents and full observability. Moreover, a range of crucial robotics-related tasks, such as multi-robot navigation and obstacle avoidance, that have been conventionally approached with the classical non-learnable methods (e.g., heuristic search) is currently suggested to be solved by the learning-based or hybrid methods. Still, in this domain, it is hard, not to say impossible, to conduct a fair comparison between classical, learning-based, and hybrid approaches due to the lack of a unified framework that supports both learning and evaluation. To this end, we introduce POGEMA, a set of comprehensive tools that includes a fast environment for learning, a generator of problem instances, the collection of pre-defined ones, a visualization toolkit, and a benchmarking tool that allows automated evaluation. We introduce and specify an evaluation protocol defining a range of domain-related metrics computed on the basics of the primary evaluation indicators (such as success rate and path length), allowing a fair multi-fold comparison. The results of such a comparison, which involves a variety of state-of-the-art MARL, search-based, and hybrid methods, are presented.

Submitted to the 38th Conference on Neural Information Processing Systems (NeurIPS 2024) Track on Datasets and Benchmarks. Do not distribute.

# 1 Introduction

Multi-agent reinforcement learning (MARL) has gained an increasing attention recently and significant progress in this field has been achieved [1, 2, 3]. MARL methods have been demonstrated to generate well-performing agents' policies in strategic games [4, 5], sport simulators [6, 7], multi-component robot control [8], city traffic control [9], and autonomous driving [10]. Currently, several ways to formulate and solve MARL problems exist, based on what information is available to the agents and what type of communication is allowed in the environment [11]. Due to the increased interest in robotic applications, decentralized cooperative learning with minimizing communication between agents has recently attracted a specific attention [12, 13]. Decentralized learning naturally suits the partial observability of the environment in which the robots usually operate. Reducing the information transmitted through the communication channels between the agents increases their degree of autonomy.

The main challenges in solving MARL problems are the non-stationarity of the multi-agent environment, the need to explicitly predict the behavior of the other agents to implement cooperative behavior, high dimensionality of the action space, which grows exponentially with the number of agents, and the sample inefficiency of existing approaches. The existing MARL including model-based and hybrid learnable methods [14, 15] exhibit faster and more stable learning in SMAC-type environments [16] with vector observations and full observability. Currently, the best results are shown by the discrete explicit world models, that use Monte Carlo tree search for planning with various heuristics to reduce the search space [17, 15].

However, in numerous practically inspired applications, like in mobile robot navigation, agents' observations are typically high-dimensional (e.g. stacked occupancy grid matrices or image-based observations as compared to 32-dim vectors in SMAC [16]) and only partially describe the state of the environment, including the other agents [18, 19]. This makes the problem specifically challenging, especially in the environments where a large number of agents are involved. For example, it is not uncommon in robotics to consider settings where up to hundreds of agents are acting (moving) simultaneously in the shared workspace as opposed to 2–10 agents in conventional MARL environments such as SMAC [16] or Google Research Football [20]. Learning to act in such crowded, observation-rich and partially-observable environments is a notable challenge to existing MARL methods.

Conventionally, the problem of multi-robot cooperative navigation (which is very important due to its applications in modern automated warehouses and fulfillment centers [21]) is framed as a search problem over a discretized search space, composed of robots-locations tuples. All robots are assumed to be confined to a graph, typically – a 4-connected grid [22], and at each time step a robot can either move following a graph's edge or stay at the current vertex. This problem setting is known as (Classical) Multi-agent Pathfinding problem [23]. Even in such simplified setting (discretized space, discretized time, uniform-duration actions etc.) obtaining a set of individual plans (one for each robot) that are mutually-conflict-free (i.e. no vertex or edge is occupied by disctinct agents at the same time step) and minimize a common objective such as, for example, the arrival time of the last agent (known as the makespan in the literature) is NP-Hard [24]. Moreover if the underlying graph is directed even obtaining a valid solution is HP-Hard as well [25].

To this end the focus of the multi-agent pathfinding community is recently being shifted towards exploring of how state-of-the-art machine learning techniques, especially reinforcement learning and imitation learning, can be leveraged to increase the efficiency of traditional solvers. Methods like [26, 27, 28, 29, 30, 31, 32, 33, 34] are all hybrid solvers that rely on both widespread search-based techniques and learnable components as well. They all are developed using different frameworks, environments and datasets and are evaluated accordingly, i.e. in the absence of the unifying evaluation framework, consisting of the (automated) evaluation tool, protocol (that defines common performance indicators) and the dataset of the problem instances. Moreover, currently most of the pure MARL methods, i.e. the ones that do not involve search-based modules, such as QMIX [35], MAMBA [14],

MAPPO [36] etc., are mostly not included in comparison. The main reason is that to train MARL policies a fast environment is needed, which is suited to cooperative multi-agent navigation.

To close the mentioned gaps we introduce POGEMA, a comprehensive set of tools that includes:

- a fast and flexible environment for learning and planning supporting several variants of the multi-robot navigation problem,

- a generator of problem instances for multi-task and generalization testing,

- a visualization toolkit to create plots for debugging and performance information and to make high-quality animations,

- a benchmarking tool that allows automated evaluation of both learnable, planning, and hybrid approaches.

Moreover, we introduce and specify an evaluation protocol defining a range of domain-related metrics computed on the basics of the primary evaluation indicators (such as success rate and path length), allowing a fair multi-fold comparison of learnable and classical methods. The results of such a comparison, which involves a range of the state-of-the-art MARL, search-based, and hybrid methods, are presented.

## 2 Related Work

Currently a huge variety of MARL environments exists that are inspired by various practical applications and encompass a broad spectrum of nuances in problem formulations. Notably, they include a diverse array of computer games [37, 16, 38, 39, 40, 41, 42, 43, 20]. Additionally, they address complex social dilemmas [44] including public goods games, resource allocation problems [45], and multi-agent coordination challenges. Some are practically inspired, showcasing tasks such as competitive object tracking [46], infrastructure management and planning [47], and automated scheduling of trains [48]. Beyond these, the environments simulate intricate, interactive systems such as traffic management and autonomous vehicle coordination [49], multi-agent control tasks [38, 50], and warehouse management [51]. Each scenario is designed to challenge and analyze the collaborative and competitive dynamics that emerge among agents in varied and complex contexts. We summarize the most wide-spread MARL environments in Table 1.

As we aim to create a lightweight and easy-to-configure multi-agent environment for reinforcement learning and pathfinding tasks, we consider the following factors essential. First and foremost, our environment is fully compatible with the native Python API: we target pure Python builds independent of hardware-specific software with a minimal number of external dependencies. Moreover, we underline the importance of constant extension and flexibility of the environment. Thus, we prioritize testing and continuous integration as cornerstones of the environment, as well as trouble-free modification of the transition dynamics. Secondly, we highlight that our environment targets generalization and may utilize procedural generation. Last but not least, we target high computational throughput (i.e., the number of environment steps per second) and robustness to an extremely large number of agents (i.e., the environment remains performant under high loads).

There are many environments inducing various types of multi-agent behaviors via different reward structures. Unfortunately, many of them require extensive Python support and rely on APIs of different programming languages (e.g., Lua, C++) for lower latency or depend on hardware-specific libraries such as XLA. Furthermore, many environments do not support generalization and lack procedural generation, especially in multi-agent cases. Additionally, customization of certain environments might be considered an issue without reverse engineering them. That's why we emphasize the superiority of the proposed benchmark.

Despite the diversity of available environments, most research papers tend to utilize only a selected few. Among these, the most popular are the StarCraft Multi-agent Challenge (SMAC), Multi-agent MuJoCo (MAMuJoCo), and Google Research Football (GRF), with SMAC being the most prevalent

Table 1: Comparison of different multi-agent reinforcement learning environments

| Environment | Repository | Navigation | Partially observable | Python based | Hardware-agnostic | Performance >10K Steps/s | Procedural generation | Requires generalization | Evaluation protocols | Tests & CI | PyPi Listed | Scalability >1000 Agents | Induced behavior |
|---|---|---|---|---|---|---|---|---|---|---|---|---|---|
| Flatland [48] | link | ✓ | ✓ | ✓ | ✗ | ✗ | ✗ | ✗ | ✓ | ✗ | ✓ | ✓ | Coop |
| GoBigger [52] | link | ✓ | ✓ | ✓ | ✓ | ✗ | ✗ | ✗ | ✓ | ✗ | ✓ | ✗ | Mixed/Coop |
| Google Research Football [20] | link | ✓ | ✓ | ✗ | ✗ | ✗ | ✗ | ✗ | ✗ | ✓ | ✗ | ✗ | Mixed |
| Griddly [53] | link | ✓ | ✓ | ✗ | ✗ | ✓ | ✓ | ✗ | ✗ | ✓ | ✓ | ✓ | Mixed |
| Hide-and-Seek [43] | link | ✓ | ✓ | ✓ | ✗ | ✗ | ✗ | ✗ | ✗ | ✗ | ✗ | ✗ | Comp |
| IMP-MARL [47] | link | ✗ | ✓ | ✓ | ✓ | ✗ | ✗ | ✗ | ✓ | ✗ | ✗ | ✗ | Coop |
| Jumanji (XLA) [42] | link | ✓ | ✓ | ✓ | ✗ | ✓ | ✗ | ✗ | ✓ | ✓ | ✓ | ✗ | Mixed |
| LBF [45] | link | ✓ | ✓ | ✓ | ✓ | ✗ | ✗ | ✗ | ✗ | ✓ | ✓ | ✗ | Coop |
| MAMuJoCo [50] | link | ✗ | ✓ | ✓ | ✓ | ✗ | ✗ | ✗ | ✗ | ✓ | ✓ | ✗ | Coop |
| MATE [46] | link | ✓ | ✓ | ✓ | ✓ | ✗ | ✗ | ✗ | ✓ | ✓ | ✗ | ✗ | Coop |
| MeltingPot [44] | link | ✓ | ✓ | ✗ | ✗ | ✗ | ✗ | ✓ | ✓ | ✓ | ✓ | ✗ | Mixed/Coop |
| Minecraft MALMO [41] | link | ✓ | ✓ | ✗ | ✗ | ✗ | ✓ | ✓ | ✓ | ✓ | ✗ | ✓ | Mixed |
| MPE [54] | link | ✓ | ✓ | ✓ | ✓ | ✓ | ✗ | ✗ | ✗ | ✗ | ✓ | ✗ | Mixed |
| MPE (XLA) [38] | link | ✓ | ✓ | ✓ | ✗ | ✓ | ✗ | ✗ | ✗ | ✓ | ✓ | ✗ | Mixed |
| Multi-agent Brax (XLA) [38] | link | ✗ | ✓ | ✓ | ✗ | ✓ | ✗ | ✗ | ✗ | ✓ | ✓ | ✗ | Coop |
| Multi-Car Racing [55] | link | ✓ | ✓ | ✓ | ✓ | ✗ | ✗ | ✗ | ✗ | ✗ | ✗ | ✗ | Comp |
| Neural MMO [40] | link | ✓ | ✓ | ✓ | ✓ | ✗ | ✓ | ✗ | ✓ | ✓ | ✓ | ✓ | Comp |
| Nocturne [49] | link | ✓ | ✓ | ✗ | ✗ | ✗ | ✗ | ✗ | ✓ | ✓ | ✗ | ✓ | Mixed |
| Overcooked [39] | link | ✓ | ✗ | ✓ | ✓ | ✗ | ✗ | ✓ | ✓ | ✓ | ✓ | ✗ | Coop |
| Overcooked (XLA) [38] | link | ✓ | ✗ | ✓ | ✗ | ✓ | ✗ | ✓ | ✗ | ✓ | ✓ | ✓ | Coop |
| RWARE [45] | link | ✓ | ✓ | ✓ | ✓ | ✓ | ✗ | ✗ | ✗ | ✓ | ✓ | ✗ | Coop |
| SISL [51] | link | ✓ | ✓ | ✓ | ✓ | ✓ | ✗ | ✗ | ✗ | ✓ | ✓ | ✗ | Coop |
| SMAC [37] | link | ✓ | ✓ | ✗ | ✗ | ✗ | ✗ | ✗ | ✓ | ✗ | ✗ | ✗ | Mixed/Coop |
| SMAC v2 [16] | link | ✓ | ✓ | ✗ | ✗ | ✗ | ✗ | ✗ | ✓ | ✗ | ✗ | ✗ | Mixed/Coop |
| SMAX (XLA) [38] | link | ✓ | ✓ | ✓ | ✗ | ✓ | ✗ | ✗ | ✗ | ✓ | ✓ | ✓ | Mixed/Coop |
| POGEMA (ours) | link | ✓ | ✓ | ✓ | ✓ | ✓ | ✓ | ✓ | ✓ | ✓ | ✓ | ✓ | Mixed |

in top conference papers. The popularity of these environments is likely due to their effective contextualization of algorithms. For instance, to demonstrate the advantages of a method, it is crucial to test it within a well-known environment.

The evaluation protocols in these environments typically feature learning curves that highlight the performance of each algorithm under specific scenarios. For SMAC, these scenarios involve games against predefined bots with specific units on both sides. In MAMuJoCo, the standard tasks involve agents controlling different sets of joints, while in GRF, the scenarios are games against predefined policies from Football Academy scenarios. Proper evaluation of MARL approaches is a serious concern. For SMAC, it's highlighted in the paper [56], which proposes a unified evaluation protocol for this benchmark. This protocol includes default evaluation parameters, performance metrics, uncertainty quantification, and a results reporting scheme.

The variability of results across different studies underscores the importance of a well-defined evaluation protocol, which should be developed alongside the presentation of the environment. In our study, we provide not only the environment but also the evaluation protocol, popular MARL baselines, and modern learnable MAPF approaches to better position our benchmark within the context.

## 3 POGEMA

POGEMA, which comes from Partially-Observable Grid Environment for Multible Agents, is an umbrella name for a collection of versatile and flexible tools aimed at developing, debugging and evaluating different methods and policies tailored to solve several types of multi-agent navigation tasks.

## 3.1 POGEMA Environment

POGEMA[1] environment is a core of POGEMA suite. It implements the basic mechanics of agents' interaction with the world. The environment can be installed using the Python Package Index (PyPI). The environemnt is open-sourced and available at github[2] under MIT license. POGEMA provides integration with existing RL frameworks: PettingZoo [57], PyMARL [58], and Gymnasium [59].

**Basic mechanics**   The workspace where the agents navigate is represented as a grid composed of blocked and free cells. Only the free cells are available for navigation. At each timestep each agent individually and independently (in accordance with a policy) picks an action and then these actions are performed simultaneously. POGEMA implements collision shielding mechanism, i.e. if an agent picks an action that leads to an obstacle (or out-of-the-map) than it stays put, the same applies for two or more agents that wish to occupy the same cell. POGEMA also has an option when one of the agents deciding to move to the common cell does it, while the others stay where they were. The episode ends when the predefined timestep, episode length, is reached. The episode can also end before this timestep if certain conditions are met, i.e. all agents reach their goal locations if MAPF problem (see below) is considered.

**Problem settings**   POGEMA supports two generic types of multi-agent navigation problems. In the first variant, dubbed MAPF (from Multi-agent Pathfinding), each agent is provided with the unique goal location and has to reach it avoiding collisions with the other agents and static obstacles. For MAPF problem setting POGEMA supports both *stay-at-target* behavior (when the episode successfully ends only if all the agents are at their targets) and *disappear-at-target* (when the agent is removed from the environment after it first reaches its goal). The second variant is a *lifelong* version of multi-agent navigation and is dubbed accordingly – LMAPF. Here each agent upon reaching a goal is immediately assigned another one (not known to the agent beforehand). Thus the agents are constantly moving trough in the environment until episode ends.

**Observation**   At each timestep each agent in POGEMA receives an individual ego-centric observation represented as a tensor – see Fig. 1. The latter is composed of the following $(2R+1) \times (2R+1)$ binary matrices, where $R$ is the observation radius set by the user:

1. Static Obstacles – 0 means the free cell, 1 – static obstacle
2. Other Agents – 0 means no agent in the cell, 1 – the other agent occupies the cell
3. Targets – projection of the (current) goal location of the agent to the boundary of its field-of-view

The suggested observation, which is, indeed, minimalist and simplistic, can be modified by the user using wrapper mechanisms. For example, it is not uncommon in the MAPF literature to augment the observation with additional matrices encoding the agent's path-to-goal (constructed by some global pathfinding routine) [27] or other variants of global guidance [29].

**Reward**   POGEMA features the most intuitive and basic reward structure for learning. I.e. an agent is rewarded with $+1$ if it reaches the goal and receives 0 otherwise. For MARL policies that leverage centralized training a shared reward is supported, i.e. $r_t = goals/agents$ where $goals$ is the number of goals reached by the agents at timestep $t$ and $agents$ is the number of agents. Indeed, the user can specify its own reward using wrappers.

**Performance indicators**   The following performance indicators are considered basic and are tracked in each episode. For MAPF they are: *Sum-of-costs* (*SoC*) and *makespan*. The former is the sum of time steps (across all agents) consumed by the agents to reach their respective goals, the latter is the maximum over those times. The lower those indicators are the more effectively the agents are solving

---

[1]https://pypi.org/project/pogema
[2]https://github.com/AIRI-Institute/pogema

MAPF tasks. For LMAPF the primary tracked indicator is the *throughput* which is the ratio of the number of the accomplished goals (by all agents) to the episode length. The higher – the better.

## 3.2 POGEMA Toolbox

The POGEMA Toolbox is a comprehensive framework designed to facilitate the testing of learning-based approaches within the POGEMA environment. This toolbox offers a unified interface that enables the seamless execution of any learnable MAPF algorithm in POGEMA. Firstly, the toolbox provides robust management tools for custom maps, allowing users to register and utilize these maps effectively within POGEMA. Secondly, it enables the concurrent execution of multiple testing instances across various algorithms in a distributed manner, leveraging Dask[3] for scalable processing. The results from these instances are then aggregated for analysis. Lastly, the toolbox includes visualization capabilities, offering a convenient method to graphically represent aggregated results through detailed plots. This functionality enhances the interpretability of outcomes, facilitating a deeper understanding of algorithm performance.

POGEMA Toolbox offers a dedicated tool for map generation, allowing the creation of three distinct types of maps: random, mazes and warehouse maps. All generators facilitates map creation using adjustable parameters such as width, height, and obstacle density. Additionally, maze generator includes specific parameters for mazes such as the number of wall components and the length of walls. The maze generator was implemented based on the generator provided in [34]. POGEMA Toolbox[4] can be installed using PyPI, and licenced under Apache License 2.0.

## 3.3 Baselines

POGEMA integrates a variety of MARL, hybrid and planning-based algorithms with the environment. These algorithms, recently presented, demonstrate state-of-the-art performance in their respective fields. Table 2 highlights the differences between these approaches. Some, such as LaCAM and RHCR, are centralized search-based planners. Other approaches, such as SCRIMP and DCC, while decentralized, still require communication between agents to resolve potential collisions. The following modern MARL algorithms are included as baselines: MAMBA [14], QPLEX [60], IQL [61], VDN [62], and QMIX [35]. For environment preprocessing, we used the preprocessing scheme provided in the Follower approach, enhancing it with the anonymous targets of other agents' local observations. We utilized the official implementation of MAMBA, as provided by its authors[5], and employed PyMARL2 framework[6] for establishing MARL baselines.

# 4 Evaluation Protocol

## 4.1 Dataset

We include the maps of the following types in our evaluation dataset (with the intuition that different maps topologies are necessary for proper assessment):

- `Mazes` – maps that encouter prolonged corridors with 1-cell width that require high level of cooperation between the agent to accomplish the mission. These maps are proceduraly generated.

- `Random` – one of the most commonly used type of maps, as they are easy to generate and allow to avoid overfitting to some special structure of the map. POGEMA ontains an integrated random maps generator, that allows to control the density of the obstacles.

- `Warehouses` – this type of maps are usually used in the papers related to LifeLong MAPF. While there is no narrow passages, high density of the agents might significantly reduce the

---

[3]https://github.com/dask/dask
[4]https://pypi.org/project/pogema-toolbox
[5]https://github.com/jbr-ai-labs/mamba
[6]https://github.com/hijkzzz/pymarl2

Table 2: This table provides an overview of various baseline approaches supported by POGEMA and their features in the context of decentralized multi-agent pathfinding.

| Algorithm | Decentralized | Partial Observability | Fully Integrated into POGEMA | Supports MAPF | Supports LifeLong MAPF | No Global Obstacles Map | No Communication | Parameter Sharing | Decentralized Learning | Model-Based | No Imitation Learning |
|---|---|---|---|---|---|---|---|---|---|---|---|
| MAMBA [14] | ✓ | ✓ | ✓ | ✓ | ✓ | ✗ | ✓ | ✓ | ✗ | ✓ | ✓ |
| QPLEX [60] | ✓ | ✓ | ✓ | ✓ | ✓ | ✗ | ✓ | ✓ | ✗ | ✗ | ✓ |
| IQL [61] | ✓ | ✓ | ✓ | ✓ | ✓ | ✗ | ✓ | ✓ | ✓ | ✗ | ✓ |
| VDN [62] | ✓ | ✓ | ✓ | ✓ | ✓ | ✗ | ✓ | ✓ | ✗ | ✗ | ✓ |
| QMIX [35] | ✓ | ✓ | ✓ | ✓ | ✓ | ✗ | ✓ | ✓ | ✗ | ✗ | ✓ |
| Follower [27] | ✓ | ✓ | ✓ | ✗ | ✓ | ✗ | ✓ | ✓ | ✓ | ✗ | ✓ |
| MATS-LP [28] | ✓ | ✓ | ✓ | ✗ | ✓ | ✗ | ✓ | ✓ | ✓ | ✓ | ✓ |
| Switcher [26] | ✓ | ✓ | ✓ | ✗ | ✓ | ✓ | ✓ | ✓ | ✓ | ✗ | ✓ |
| SCRIMP [30] | ✓ | ✓ | ✗ | ✓ | ✗ | ✗ | ✗ | ✓ | ✗ | ✗ | ✗ |
| DCC [29] | ✓ | ✓ | ✓ | ✓ | ✗ | ✗ | ✗ | ✓ | ✗ | ✗ | ✗ |
| LaCAM [63] | ✗ | ✗ | ✗ | ✓ | ✗ | ✗ | - | - | - | - | - |
| RHCR [64] | ✗ | ✗ | ✗ | ✗ | ✓ | ✗ | - | - | - | - | - |

overall throughput, especially when agents are badly distributed along the map. These maps are also can be produraly generated.

- MovingAI – a set of maps from the existing benchmark widely used in MAPF community. The contained maps have different sizes and structures. It can be used to show how the approach deals with single-agent pathfinding and also deals with the maps that have out-of-distribution structure.

- MovingAI-tiles – a modified MovingAI set of maps. Due to the large size of the original maps, it's hard to get high density of the agents on them. To get more crowded maps, we slice the original maps on 16 pieces with $64 \times 64$ size.

- Puzzles – a set of small hand-crafted maps that contains some difficult patterns that mandate the cooperation between that agents.

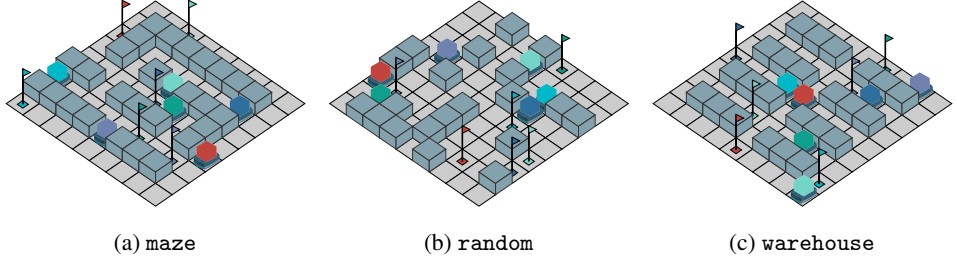

(a) maze         (b) random         (c) warehouse

Figure 2: Examples of maps presented in POGEMA.

Start and goal locations are generated via random generators. They are generated with fixed seeds, thus can be reproduced. It's guaranteed, that each agent has its own goal location and the path to it from its start location exists.

## 4.2 Metrics

The existing works related to solving MAPF problems evaluates the performance by two major criteria – success rate and the primary performance indicators mentioned above: sum-of-costs, makespan, throughput. These are directly obtainable from POGEMA. While these metrics allow to evaluate the algorithms at some particular instance, it's might be difficult to get a high-level conclusion about the

performance of the algorithms. Thus, we want to introduce several high-level metrics that covers multiple different aspects:

`Performance` – how well the algorithm works compared to other approaches. To compute this metric we run the approaches on a set of maps similar to the ones, used during training, and compare the obtained results with the best ones.

$$Performance_{MAPF} = \begin{cases} SoC_{best}/SoC \\ 0 \text{ if not solved} \end{cases} \tag{1}$$

$$Performance_{LMAPF} = throughput/throughput_{best} \tag{2}$$

`Out-of-Distribution` – how well the algorithm works on out-of-distribution maps. This metric is computed in the same way as `Performance`, with the only difference that the approaches are evaluated on a set of maps, that were not used during training phase and have different structure of obstacles. For this purpose we utilize maps from `MovingAI-tiles` set of maps.

$$Out\_of\_Distribution_{MAPF} = \begin{cases} SoC_{best}/SoC \\ 0 \text{ if not solved} \end{cases} \tag{3}$$

$$Out\_of\_Distributionn_{LMAPF} = throughput/throughput_{best} \tag{4}$$

`Scalability` – how well the algorithm scales to large number of agents. To evaluate how well the algorithm scales to large number of agents, we run it on a large warehouse map with increasing number of agents and compute the ratio between runtimes with various number of agents.

$$Scalability = \frac{runtime(agents_1)/runtime(agents_2)}{|agents_1|/|agents_2|} \tag{5}$$

`Cooperation` – how well the algorithm is able to resolve complex situations. To evaluate this metric we run the algorithm on `Puzzles` set of maps and compare the obtained results with best solutions that were obtained by classical MAPF/LMAPF solvers.

$$Cooperation_{MAPF} = \begin{cases} SoC_{best}/SoC \\ 0 \text{ if not solved} \end{cases} \tag{6}$$

$$Cooperation_{LMAPF} = throughput/throughput_{best} \tag{7}$$

`Congestion` – how well the algorithm distributes the agents along the map and reduces redundant waits, collisions, etc. To evaluate this metric we compute the average density of the agents presented in the observations of each agent and compare it to the overall density of the agents on the map.

$$Congestion = \frac{\sum_{i \in agents} agents\_density(obs_i)/agents\_density(map)}{|agents|} \tag{8}$$

`Pathfinding` – how well the algorithm works in case of presence of a single agent on a large map. This metric is tailored to determine the ability of the approach to effectively lead agents to their goal locations. For this purpose we run the approaches on large city maps from `MovingAI` benchmark sets. The obtained solution cost (in fact - length of the path) should be optimal.

$$Pathfinding = \begin{cases} 1 \text{ if path is optimal} \\ 0 \text{ otherwise} \end{cases} \tag{9}$$

### 4.3 Experimental Results

We have evaluated a bunch of the algorithms on both MAPF and LMAPF setups on all 6 datasets. The results of this evaluation are presented in Fig.3. The details about number of maps, number of agents, seeds, etc. are given in the supplementary material (as well as details on how these results can be reproduced).

In both setups, i.e. MAPF and LMAPF, the best results in terms of cooperation, out-of-distribution and performance metrics were obtained by centralized planners, i.e. LaCAM and RHCR respectively.

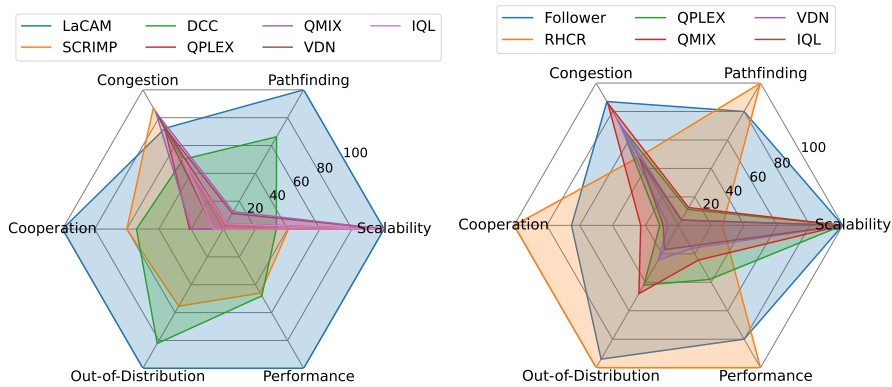

Figure 3: Evaluation of baselines available in POGEMA on (a) MAPF (b) LMAPF instances.

For MAPF tasks, LaCAM outperformed all other approaches on all metrics except congestion. It is hypothesized that in this approach, the even distribution of agents across the environment is not crucial due to its centralized nature, which efficiently resolves complex conflicts. Specialized learnable MAPF approaches, i.e., DCC and SCRIMP, take second place, showing close performance but with different specifics. DCC shows better results on out-of-distribution tasks and pathfinding tasks than SCRIMP, which is better at managing congestion. Surprisingly, the results of SCRIMP are inferior on pathfinding tasks, suggesting a problem with this approach in single-agent tasks that do not require communication, which can be an out-of-distribution setup for this algorithm. MARL algorithms such as QPLEX, VDN, and QMIX underperform in comparison with other approaches, exhibiting a significant gap in the results, which can be attributed to the absence of additional techniques used in hybrid approaches, despite incorporating preprocessing techniques from the Follower approach. This could suggest that the MARL community lacks large-scale approaches and benchmarks for them. Predictably, IQL shows the poorest performance, highlighting the importance of centralized training for multi-agent pathfinding (MAPF) tasks that require high levels of cooperation.

For LMAPF, the situation changes dramatically. The centralized approach, RHCR, dominates in cooperation, out-of-distribution tasks, and overall performance. However, it significantly lags behind Follower in terms of congestion and scalability metrics. The superior performance of Follower can be attributed to a dedicated technique tailored to avoid congestion. The most crucial metric here is performance, where Follower outperforms RHCR by a considerable margin, while not underperforming significantly in cooperation, out-of-distribution tasks, and pathfinding metrics. This showcases how applying learnable methods can substantially enhance the applicability of these approaches. Additionally, the high performance of Follower can be linked to large-scale training setups, including billions of training steps. Again, MARL approaches underperform in these scenarios, with QMIX and QPLEX showing comparable results. QMIX performs better in cooperation and out-of-distribution metrics, while QPLEX excels in performance.

## 5 Conclusion and Limitations

This paper presents POGEMA – a powerful suite of tools tailored for creating, assessing, and comparing methods and policies in multi-agent navigation problems. POGEMA encompasses a fast learning environment and a comprehensive evaluation toolbox suitable for pure MARL, hybrid, and search-based solvers. It includes a wide array of methods as baselines. The evaluation protocol described, along with a rich set of metrics, assists in assessing the generalization and scalability of all approaches. Visualization tools enable qualitative examination of algorithm performance. Integration with the well-known MARL API and map sets facilitates the benchmark's expansion. Existing limitations are two-fold. First, a conceptual limitation is that communication between the agents is not currently disentangled in POGEMA environment. Second, the technical limitations include the lack of Jax support and integration with other well-known GPU parallelization tools.

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
