# POGEMA: A Benchmark Platform for Cooperative Multi-Agent Navigation (Supplementary material)

**Alexey Skrynnik**
AIRI
Moscow, Russia

**Anton Andreychuk**
AIRI
Moscow, Russia

**Anatolii Borzilov**
MIPT
Moscow, Russia

**Alexander Chernyavskiy**
MIPT
Moscow, Russia

**Konstantin Yakovlev**
FRC CSC RAS, AIRI
Moscow, Russia

**Aleksandr Panov**
AIRI, MIPT
Moscow, Russia

## 1 Extended Evaluation Results

POGEMA benchmark contains 6 different sets of maps and all baseline approaches were evaluated on them either on MAPF or on LMAPF instances. Regardless the type of instances, number of maps, seeds and agents were the same. Table 1 contains all information about these numbers. Note that there is no MaxSteps (LMAPF) value for `MovingAI` set of maps. This set of maps was used only for pathfinding meta-metric, thus all approaches were evaluated only on MAPF instances with a single agent. The source code for POGEMA Benchmark is available at link[1].

Table 1: Details about the instances on different sets of maps.

|  | Agents | Maps | MapSize | Seeds | MaxSteps (MAPF) | MaxSteps (LMAPF) |
|---|---|---|---|---|---|---|
| Random | [8, 16, 24, 32, 48, 64] | 128 | 17×17 - 21×21 | 1 | 128 | 256 |
| Mazes | [8, 16, 24, 32, 48, 64] | 128 | 17×17 - 21×21 | 1 | 128 | 256 |
| Warehouse | [32, 64, 96, 128, 160, 192] | 1 | 33×46 | 128 | 128 | 256 |
| Puzzles | [2, 3, 4] | 16 | 5×5 | 10 | 128 | 256 |
| MovingAI | [1] | 8 | 256×256 | 10 | 2048 | - |
| MovingAI-tiles | [64, 128, 192, 256] | 128 | 64×64 | 1 | 256 | 256 |

### 1.1 MAPF Benchmark: Performance

The performance metrics were calculated using `Mazes` and `Random` maps of size close to $20 \times 20$. The primary metrics here are SoC and CSR. The results of all the MAPF approaches over different numbers of agents are presented in Figure 1. The superior performance is shown by the centralized approach, LaCAM. The learnable approaches, DCC and SCRIMP, show comparable results. Interestingly, the former has a better SoC metric, despite the latter having better results on CSR. Among the MARL methods, better results are shown by MAMBA for both metrics.

### 1.2 MAPF Benchmark: Out-of-Distribution

Out-of-Distribution metric was calculated on `MovingAI-tiles` dataset, that consists of pieces of cities maps with $64 \times 64$ size. Due to much larger size compared to `Mazes` and `Random` maps, the amount of agents was also significantly increased. Here again centralized search-based planner, i.e.

---

[1]https://github.com/Tviskaron/pogema_benchmark

Submitted to the 38th Conference on Neural Information Processing Systems (NeurIPS 2024) Track on Datasets and Benchmarks. Do not distribute.

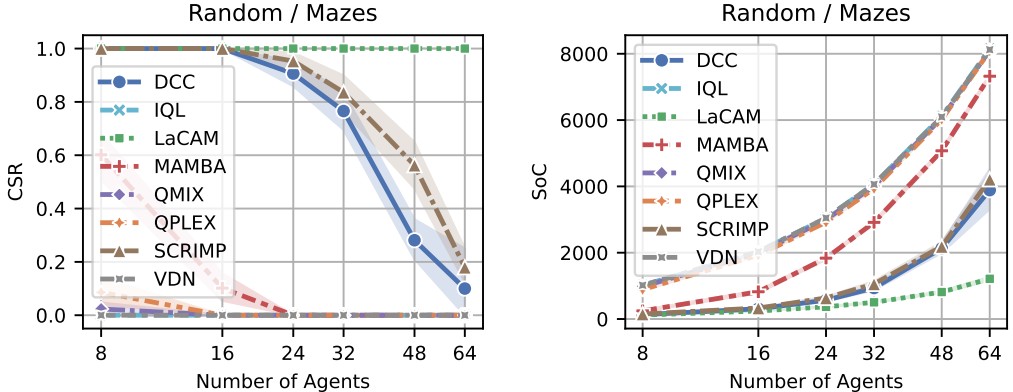

Figure 1: Plots show performance of MAPF approaches on `Random` and `Mazes` maps. The performance metrics was calculated based on SoC (lower is better) and CSR (higher is better) metrics. The shaded area indicates 95% confidence intervals.

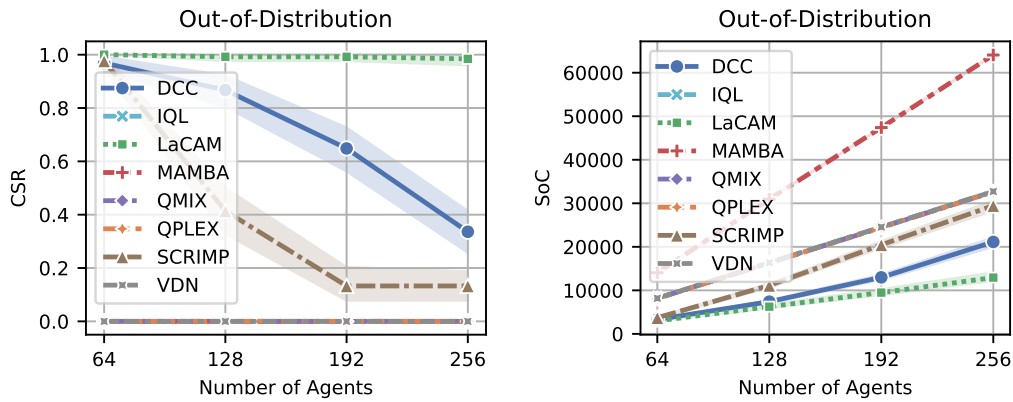

Figure 2: Plots show performance of MAPF approaches on `MovingAI-tiles` maps. These results were utilized to compute Out-of-Distribution metric. The shaded area indicates 95% confidence intervals.

LaCAM, demonstrates the best results both in terms of CSR and SoC. Hybrid methods methods, i.e. DCC and SCRIMP, are also able to solve some of the instances. While DCC and SCRIMP demonstrate similar results on `Mazes` and `Random` maps, DCC completely outperforms SCRIMP on out-of-distribution dataset. MARL approaches are not able to solve any instance even with 64 agents.

## 1.3 MAPF Benchmark: Scalability

The results of how well the algorithm scales with a large number of agents are shown in Figure 3. The experiments were conducted on a `warehouse` map. The plot is log-scaled. The best scalability is achieved with the centralized LaCAM approach, which is a high-performance approach. The worst results in both runtime and scalability are for SCRIMP, with results close to it for DCC. Despite an initially high runtime, the scalability of MAMBA is better than other approaches; however, this could be attributed to the high cost of GPU computation, which is due to the large number of parameters in the neural network and is the limiting factor of this approach.

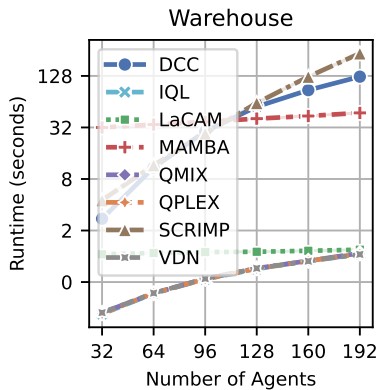

Figure 3: Runtime in seconds for each algorithm. Note that the plot is log-scaled.

### 1.4 MAPF Benchmark: Cooperation

How well the algorithm is able to resolve complex situations on the `Puzzles` set is reflected in the results presented in Table 2. Surprisingly, the centralized approach LaCAM does not solve all the tasks, showing only a 0.96 CSR. This highlights that this type of task is difficult even for centralized approaches, despite the small map size of $5 \times 5$ and the low number of agents $(2 - 4)$. SCRIMP outperformed DCC in CSR but again showed comparable results in SoC. Among MARL approaches, better cooperation is demonstrated by QMIX, outperforming QPLEX, VDN, IQL, and even MAMBA.

### 1.5 MAPF Benchmark: Congestion

Congestion is one of the meta-metrics that estimates how well the agents are distributed along the map. This metric indirectly influences the performance of the approach, as well-distributed agents in case of highly crowded instances allows to reduce the amount of collisions and redundant wait-actions. To compute this metric we utilize the results obtained on `Warehouse` map with the highest evaluated amount of agents - 192. In contrast to other metrics, that are computed as a ratio to the best obtained results, Congestion metric is computed as the ratio to average density of all agents on the map to average density of the agents in agents' local observations.

Table 2: Comparison of algorithms cooperation on `Puzzles` set. $\pm$ shows confidence intervals $95\%$.

| Algorithm | CSR | SoC |
|-----------|-----|-----|
| DCC | $0.72 \pm 0.04$ | $96.33 \pm 9.97$ |
| IQL | $0.27 \pm 0.04$ | $316.87 \pm 14.47$ |
| LaCAM | $0.96 \pm 0.02$ | $36.29 \pm 7.26$ |
| MAMBA | $0.39 \pm 0.04$ | $177.64 \pm 14.68$ |
| QMIX | $0.53 \pm 0.05$ | $246.11 \pm 17.10$ |
| QPLEX | $0.33 \pm 0.04$ | $250.12 \pm 14.80$ |
| SCRIMP | $0.82 \pm 0.04$ | $104.31 \pm 13.73$ |
| VDN | $0.11 \pm 0.03$ | $336.99 \pm 12.45$ |

Table 3: Average agents density by number of agents across algorithms for `Warehouse` map. Please note, only column with 192 agents was utilized to compute Congestion metric.

| Algorithm | 64 Agents | 96 Agents | 128 Agents | 160 Agents | 192 Agents |
|-----------|-----------|-----------|------------|------------|------------|
| DCC | $0.094 \pm 0.001$ | $0.132 \pm 0.001$ | $0.170 \pm 0.001$ | $0.207 \pm 0.001$ | $0.241 \pm 0.001$ |
| IQL | $0.076 \pm 0.001$ | $0.114 \pm 0.001$ | $0.163 \pm 0.002$ | $0.244 \pm 0.002$ | $0.299 \pm 0.002$ |
| LaCAM | $0.082 \pm 0.001$ | $0.116 \pm 0.001$ | $0.149 \pm 0.001$ | $0.179 \pm 0.001$ | $0.207 \pm 0.001$ |
| MAMBA | $0.101 \pm 0.001$ | $0.183 \pm 0.001$ | $0.266 \pm 0.002$ | $0.335 \pm 0.002$ | $0.389 \pm 0.002$ |
| QMIX | $0.073 \pm 0.001$ | $0.103 \pm 0.001$ | $0.130 \pm 0.001$ | $0.154 \pm 0.001$ | $0.179 \pm 0.001$ |
| QPLEX | $0.077 \pm 0.001$ | $0.113 \pm 0.001$ | $0.146 \pm 0.001$ | $0.175 \pm 0.001$ | $0.205 \pm 0.001$ |
| SCRIMP | $0.074 \pm 0.001$ | $0.104 \pm 0.001$ | $0.127 \pm 0.001$ | $0.148 \pm 0.001$ | $0.173 \pm 0.001$ |
| VDN | $0.071 \pm 0.001$ | $0.101 \pm 0.001$ | $0.130 \pm 0.001$ | $0.158 \pm 0.001$ | $0.188 \pm 0.001$ |

### 1.6 MAPF Benchmark: Pathfinding

To compute Pathfidning metric we run the approaches on the instances with a single agent. For this purpose we utilized large `MovingAI` mapf with $256 \times 256$ size. While this task seems easy, most of the hybrid and MARL approaches are not able to effectively solve them. Only LaCAM is able to find optimal paths in all the cases, as it utilizes precomputed costs to the goal location as a heuristic. Most of the evaluated hybrid and MARL approaches are also contain a sort of global guidance in one the channels of their observations. However, large maps with out-of-distribution structure, the absence of communication and other agents in local observations are able to lead to inconsistent behavior of the models that are not able

Table 4: Comparison of makespan used for pathfinding metric.

| Algorithm | Makespan |
|-----------|----------|
| DCC | $189.56 \pm 28.28$ |
| IQL | $1825.95 \pm 137.11$ |
| LaCAM | $179.82 \pm 20.21$ |
| MAMBA | $416.45 \pm 139.34$ |
| QMIX | $955.54 \pm 203.76$ |
| QPLEX | $933.74 \pm 204.21$ |
| SCRIMP | $1460.04 \pm 176.27$ |
| VDN | $1733.20 \pm 158.91$ |

72 to effectively choose the actions that lead the agent to its goal. Please note, SoC and makespan
73 metrics in this case are equal, as there is only one agent in every instance.

## 1.7 LifeLong MAPF Benchmark: Performance

75 Performance metric in LMAPF case is based on the ratio of throughput compared to the best obtained
76 one. In contrast to SoC, throughput should be as high as possible. There is also no CSR metric,
77 as there is no need for agents to be at their goal locations simultaneously. As well as in MAPF
78 case, the best results are obtained by centralized search-based approach – RHCR. The best results
79 among decentralized methods demonstrate Follower and MATS-LP. Between pure MARL methods
80 the highest throughput on both `Random` and `Mazes` maps is obtained by MAMBA. The one can also
81 note multiple approaches, that were not directly mentioned in the baselines section – ASwitcher,
82 HSwitcher, LSwitcher, EPOM and RePlan. All these approaches are parts of Switcher baseline, where
83 RePlan is search-based planner, EPOM – learn-based, and the rest are the switchers that combine
84 these two methods. Between them the best results demonstrates ASwitcher.

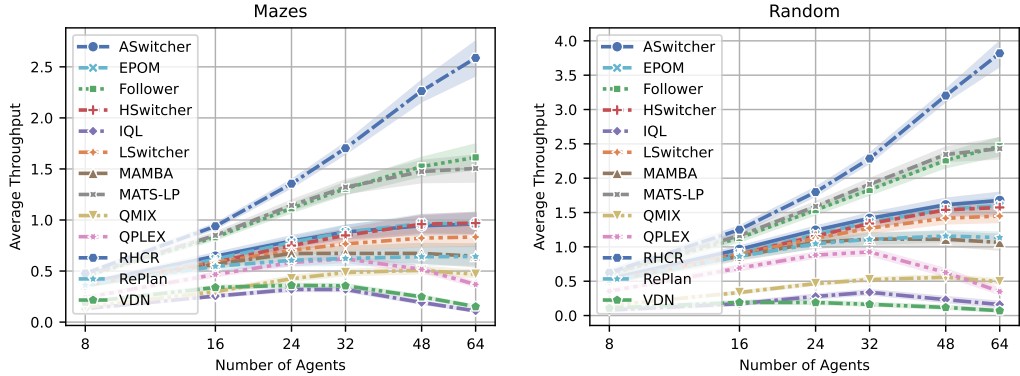

Figure 4: Performance results for LifeLong scenarios on the `Mazes` and `Random` maps.

## 1.8 LifeLong MAPF Benchmark: Out-of-Distribution

86 The evaluation on out-of-distribution set of maps confirms the results obtained on `Random` and `Mazes`
87 maps. The best results demonstrates RHCR. Next best results are obtained by Follower and MATS-LP,
88 which are much closer to RHCR in this experiment. While MATS-LP outperforms Follower on the
89 instances with 64, 128 and 192 agents, Follower is still better on the instances with 256 agents. Such
90 relation is probably explained by the presence of dynamic edge-costs in Follower that allows to better
91 distribute agents along the map and reduce congestion between them.

## 1.9 LifeLong MAPF Benchmark: Scalability

93 Figure 5 contains log-scaled plot of average time spent
94 by each of the algorithms to process an instance on
95 `Warehouse` map with the corresponding amount of agents.
96 Most of the approaches scales almost linearly, except
97 RHCR. This centralized search-based method lacks of
98 exponential grow, as it needs to find a collision-free solu-
99 tion for at least next few steps, rather than just to make
100 a decision about next action for each of the agents. The
101 worst runtime demonstrate MATS-LP, as it runs MCTS
102 and simulates the behavior of the other observable agents.
103 It's still scales better than RHCR as it builds trees for each
104 of the agents independently.

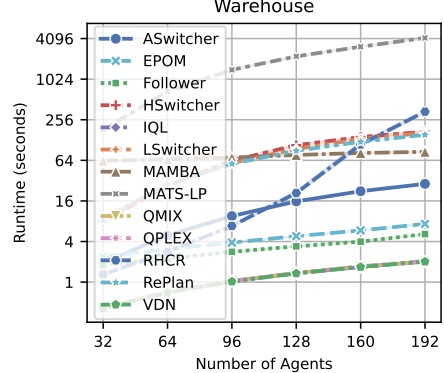

Figure 5: Runtime in seconds for each al-
gorithm. Note that the plot is log-scaled.

Table 5: Evaluation on Out-of-Distribution maps. The results are shown by average throughput metric.

| Algorithm | 64 Agents | 128 Agents | 192 Agents | 256 Agents |
|---|---|---|---|---|
| ASwitcher | $1.26 \pm 0.08$ | $2.30 \pm 0.13$ | $3.14 \pm 0.17$ | $3.80 \pm 0.20$ |
| EPOM | $1.18 \pm 0.08$ | $2.19 \pm 0.13$ | $3.01 \pm 0.17$ | $3.60 \pm 0.20$ |
| Follower | $1.50 \pm 0.08$ | $2.82 \pm 0.13$ | $3.95 \pm 0.19$ | $4.81 \pm 0.22$ |
| HSwitcher | $1.24 \pm 0.08$ | $2.22 \pm 0.12$ | $3.01 \pm 0.17$ | $3.58 \pm 0.20$ |
| IQL | $0.26 \pm 0.02$ | $0.65 \pm 0.04$ | $0.78 \pm 0.05$ | $0.68 \pm 0.05$ |
| LSwitcher | $1.23 \pm 0.07$ | $2.23 \pm 0.12$ | $3.06 \pm 0.17$ | $3.67 \pm 0.20$ |
| MAMBA | $1.02 \pm 0.05$ | $1.42 \pm 0.08$ | $2.05 \pm 0.12$ | $2.46 \pm 0.17$ |
| MATS-LP | $1.57 \pm 0.12$ | $2.98 \pm 0.20$ | $4.04 \pm 0.33$ | $4.69 \pm 0.39$ |
| QMIX | $0.83 \pm 0.03$ | $1.55 \pm 0.06$ | $2.01 \pm 0.09$ | $2.27 \pm 0.11$ |
| QPLEX | $0.79 \pm 0.03$ | $1.48 \pm 0.07$ | $1.79 \pm 0.10$ | $1.74 \pm 0.11$ |
| RHCR | $1.57 \pm 0.08$ | $3.00 \pm 0.14$ | $4.22 \pm 0.23$ | $5.13 \pm 0.34$ |
| RePlan | $1.24 \pm 0.08$ | $2.15 \pm 0.12$ | $2.82 \pm 0.17$ | $3.25 \pm 0.19$ |
| VDN | $0.50 \pm 0.03$ | $0.91 \pm 0.05$ | $1.03 \pm 0.06$ | $0.96 \pm 0.06$ |

## 1.10 LifeLong MAPF Benchmark: Cooperation

As well as for MAPF setting, cooperation metric is computed based on the results obtained on `Puzzles` dataset. Table 6 contains average throughput obtained by each of the evaluated approaches. Here again the best results are obtained by RHCR algorithm. In contrast to `Random`, `Mazes` and `Warehouse` sets of maps, where MATS-LP and Follower demonstrate close results, the ability to simulate the behavior of other agents, provided by MCTS in MATS-LP, allows to significantly outperform Follower on small `Puzzles` maps. The rest approaches demonstrate much worse results, especially IQL, QPLEX and VDN that have 10 times worse average throughput than RHCR.

## 1.11 LifeLong MAPF Benchmark: Congestion

Table 7 contains average agents density presented in observations. As it was already mentioned in LMAPF:Out-of-Distribution section, Follower contains a mechanism that allows to effectively distribute agents along the map. As a result, the lowest density is demonstrated exactly by this approach. MARL methods, such as MAMBA and QMIX are also demonstrate low average agents density, as they actually utilize the same observation as Follower does.

Table 6: Average throughput on `Puzzles` maps that were used to compute Cooperation metric.

| Algorithm | Average Throughput |
|---|---|
| ASwitcher | $0.164 \pm 0.015$ |
| EPOM | $0.147 \pm 0.014$ |
| Follower | $0.319 \pm 0.020$ |
| HSwitcher | $0.194 \pm 0.014$ |
| IQL | $0.036 \pm 0.003$ |
| LSwitcher | $0.206 \pm 0.013$ |
| MAMBA | $0.133 \pm 0.014$ |
| MATS-LP | $0.394 \pm 0.021$ |
| QMIX | $0.117 \pm 0.010$ |
| QPLEX | $0.051 \pm 0.006$ |
| RHCR | $0.538 \pm 0.021$ |
| RePlan | $0.194 \pm 0.013$ |
| VDN | $0.030 \pm 0.004$ |

Table 7: Average Agents Density by Number of Agents

| Algorithm | 64 Agents | 96 Agents | 128 Agents | 160 Agents | 192 Agents |
|---|---|---|---|---|---|
| ASwitcher | $0.074 \pm 0.001$ | $0.111 \pm 0.001$ | $0.146 \pm 0.001$ | $0.176 \pm 0.001$ | $0.203 \pm 0.001$ |
| EPOM | $0.075 \pm 0.001$ | $0.122 \pm 0.002$ | $0.180 \pm 0.003$ | $0.230 \pm 0.003$ | $0.269 \pm 0.003$ |
| Follower | $0.073$ | $0.101$ | $0.126$ | $0.150$ | $0.173$ |
| HSwitcher | $0.075 \pm 0.001$ | $0.121 \pm 0.002$ | $0.176 \pm 0.003$ | $0.227 \pm 0.003$ | $0.270 \pm 0.003$ |
| IQL | $0.080 \pm 0.001$ | $0.127 \pm 0.002$ | $0.188 \pm 0.003$ | $0.257 \pm 0.003$ | $0.319 \pm 0.002$ |
| LSwitcher | $0.075 \pm 0.001$ | $0.114 \pm 0.001$ | $0.149 \pm 0.001$ | $0.180 \pm 0.001$ | $0.208 \pm 0.001$ |
| MAMBA | $0.073$ | $0.095$ | $0.119 \pm 0.001$ | $0.146 \pm 0.001$ | $0.176 \pm 0.001$ |
| MATS-LP | $0.110 \pm 0.002$ | $0.176 \pm 0.002$ | $0.231 \pm 0.003$ | $0.274 \pm 0.003$ | $0.306 \pm 0.003$ |
| QMIX | $0.074$ | $0.100$ | $0.126$ | $0.150 \pm 0.001$ | $0.175 \pm 0.001$ |
| QPLEX | $0.078$ | $0.114 \pm 0.001$ | $0.147 \pm 0.001$ | $0.176 \pm 0.001$ | $0.216 \pm 0.002$ |
| RHCR | $0.088$ | $0.125 \pm 0.001$ | $0.170 \pm 0.002$ | $0.242 \pm 0.004$ | $0.314 \pm 0.004$ |
| RePlan | $0.081 \pm 0.001$ | $0.116 \pm 0.001$ | $0.145 \pm 0.001$ | $0.168 \pm 0.001$ | $0.189 \pm 0.001$ |
| VDN | $0.077 \pm 0.001$ | $0.109 \pm 0.001$ | $0.141 \pm 0.001$ | $0.173 \pm 0.001$ | $0.204 \pm 0.002$ |

## 1.12 LifeLong MAPF Benchmark: Pathfinding

Pathfinding metric is tailored to indicate how well the algorithm is able to guide am agent to its goal location. As a result, there is actually no need to run the algorithms on LifeLong instances. Instead, they were run on the same set of instances that were utilized for MAPF approaches. The results of this evaluation are presented in Table 8. Again, the best results were obtained by search-based approach – RHCR. Its implementation was slightly modified to work on MAPF instances, when there is no new goal after reaching the current one. Either optimal or close to optimal paths are able to find Follower and MATS-LP. Followers misses optimal paths due to the integrated technique that changes the edge-costs. MATS-LP adds noise to the root of the search tree that might result in choosing of wrong actions. For the approaches from Switcher family it's actually almost impossible to find optimal paths as they have no information about global map and operate only based on the local observations.

Table 8: Pathfinding results.

| Algorithm | Makespan |
| --- | --- |
| ASwitcher | 340.56±79.41 |
| EPOM | 762.94±168.21 |
| Follower | 181.00±20.95 |
| HSwitcher | 299.90±62.73 |
| IQL | 1825.95±144.16 |
| LSwitcher | 472.64±119.23 |
| MAMBA | 416.45±136.01 |
| MATS-LP | 179.93±22.45 |
| QMIX | 955.54±200.68 |
| QPLEX | 933.74±199.18 |
| RHCR | 179.82±20.21 |
| RePlan | 299.90±62.40 |
| VDN | 1733.20±157.96 |

# 2 Code examples for POGEMA

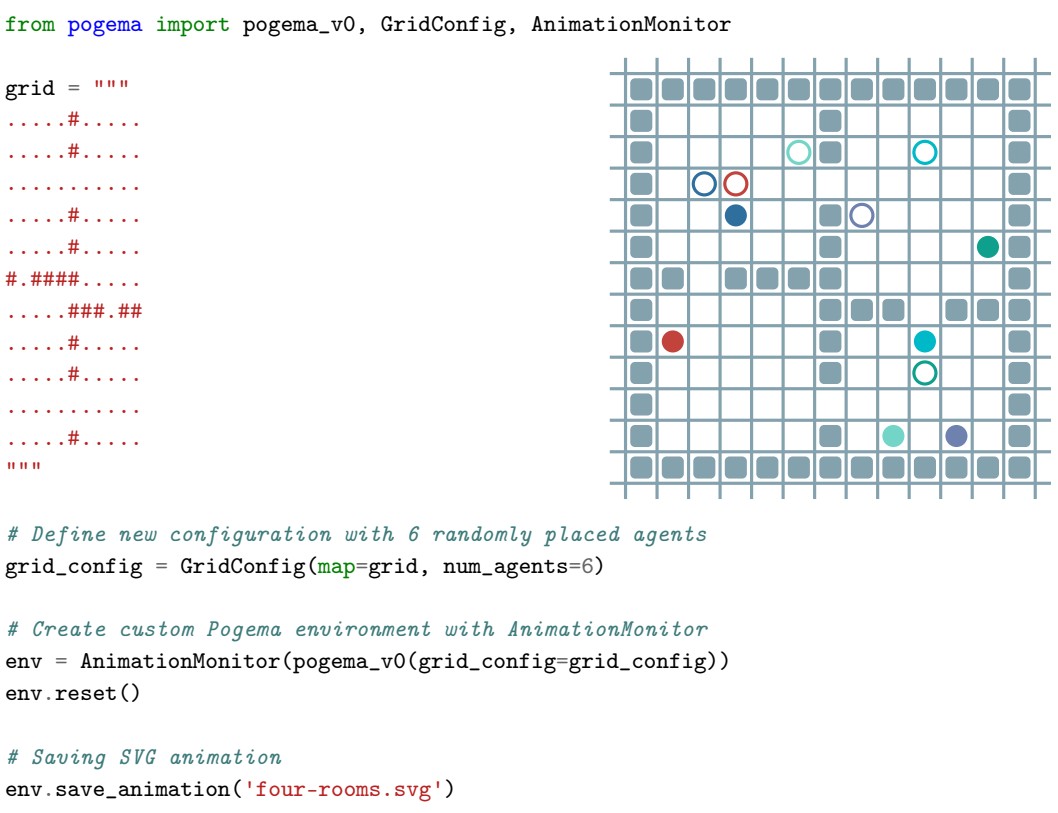

```python
from pogema import pogema_v0, GridConfig, AnimationMonitor

grid = """
.....#.....
.....#.....
...........
.....#.....
.....#.....
#.####.....
.....###.##
.....#.....
.....#.....
...........
.....#.....
"""

# Define new configuration with 6 randomly placed agents
grid_config = GridConfig(map=grid, num_agents=6)

# Create custom Pogema environment with AnimationMonitor
env = AnimationMonitor(pogema_v0(grid_config=grid_config))
env.reset()

# Saving SVG animation
env.save_animation('four-rooms.svg')
```

Listing 1: Setting up a POGEMA instance with a custom map and generating an animation.

POGEMA is an environment that provides a simple scheme for creating MAPF scenarios, specifying the parameters of `GridConfig`. The main parameters are: `on_target` (the behavior of an agent on

the target, e.g., *restart* for LifeLong MAPF and *nothing* for classical MAPF), `seed` – to preserve the same generation of the map; agent; and their targets for scenario, `size` – used for cases without custom maps to specify the size of the map, `density` – the density of obstacles, `num_agents` – the number of agents, `obs_radius` – observation radius, `collision_system` – controls how conflicts are handled in the environment (we used a *soft* collision system for all of our experiments). The example of creation such instance is presenten in Listing 1.

Visualization of the agents is a crucial tool for debugging algorithms, visually comparing them, and presenting the results. Many existing MARL environments lack such tools, or have limited visualization functionality, e.g., requiring running the simulator to provide replays, or offering visualizations only in one format (such as videos). In the POGEMA environment, there are three types of visualization formats. The first one is console rendering, which can be used with the default `render` methods of the environment; this approach is useful for local or server-side debugging. The preferred second option is *SVG* animations. An example of generating such a visualization is presented in the listing above. This approach allows displaying the results using any browser. It provides the ability to highlight high-quality static images (e.g., as the images provided in the paper) or to display results on a website (e. g., animations of the POGEMA repository on GitHub). This format ensures high-quality vector graphics. The third option is to render the results to video format, which is useful for presentations and videos.

## 3 POGEMA Toolbox

The POGEMA Toolbox provides three types of functionality.

The first one is registries to handle custom maps and algorithms. To create a custom map, the user first needs to define it using ASCII symbols or by uploading it from a file, and then register it using the toolbox (see Listing 1). The same approach is used to register and create algorithms (see Listing 2). In that listing, the registration of a simple algorithm is presented, which must includ two methods: `act` and `reset_states`. This approach can also accommodate a set of hyperparameters which the Toolbox handles.

```python
from pogema import BatchAStarAgent

# Registring A* algorithm
ToolboxRegistry.register_algorithm('A*', BatchAStarAgent)

# Creating algorithm
algo = ToolboxRegistry.create_algorithm("A*")
```

Listing 2: Example of registering the A* algorithm as an approach in the POGEMA Toolbox.

```python
from pogema_toolbox.registry import ToolboxRegistry

# Creating cusom_map
custom_map = """
.......#.
...#...#.
.#.###.#.
"""

# Registring custom_map
ToolboxRegistry.register_maps({"custom_map": custom_map})
```

Listing 3: Example of registering a custom map to the Pogema Toolbox.

```
environment: # Configuring Test Environments
  name: Environment
  on_target: 'restart'
  max_episode_steps: 128
  observation_type: 'POMAPF'
  collision_system: 'soft'
  seed:
    grid_search: [ 0, 1, 2, 3, 4, 5, 6, 7, 8, 9 ]
  num_agents:
    grid_search: [ 8, 16, 24, 32, 48, 64 ]
  map_name:
    grid_search: [
        validation-mazes-seed-000, validation-mazes-seed-001, validation-mazes-seed-002,
        validation-mazes-seed-003, validation-mazes-seed-004, validation-mazes-seed-005,
    ]

algorithms: # Specifying algorithms and it's hyperparameters
  RHCR-5-10:
    name: RHCR
    parallel_backend: 'balanced_dask'
    num_process: 32
    simulation_window: 5
    planning_window: 10
    time_limit: 10
    low_level_planner: 'SIPP'
    solver: 'PBS'

results_views: # Defining results visualization
  01-mazes:
    type: plot
    x: num_agents
    y: avg_throughput
    width: 4.0
    height: 3.1
    line_width: 2
    use_log_scale_x: True
    legend_font_size: 8
    font_size: 8
    name: Mazes
    ticks: [8, 16, 24, 32, 48, 64]

  TabularThroughput:
    type: tabular
    drop_keys: [ seed, map_name]
    print_results: True
```

Listing 4: Example of the POGEMA Toolbox configuration for parallel testing of the RHCR approach and visualization of its results.

Second, it provides a unified way of conducting distributed testing using Dask [2] and defined configurations. An example of such a configuration is provided in Listing 4. The configuration is split

---

[2]https://github.com/dask/dask

into three main sections; the first one details the parameters of the POGEMA environment used for testing. It also includes iteration over the number of agents, seeds, and names of the map (which were registered beforehand). The unified `grid_search` tag allows for the examination of any existing parameter of the environment. The second part of the configurations is a list of algorithms to be tested. Each algorithm has its alias (which will be shown in the results) and name, which specifies the family of methods. It also includes a list of hyperparameters common to different approaches, e.g., number of processes, parallel backend, etc., and the specific parameters of the algorithm.

The third functionality and third part of the configuration concern views. This is a form of presenting the results of the algorithms. Working with complex testing often requires custom tools for creating visual materials such as plots and tables. The POGEMA toolbox provides such functionality for MAPF tasks out-of-the-box. The listing provides two examples of such data visualization: a plot and a table, which, based on the configuration, provide aggregations of results and present information in a high-quality form, including confidence intervals. The plots and tables in the paper are prepared using this functionality.

# 4 Examples of used maps

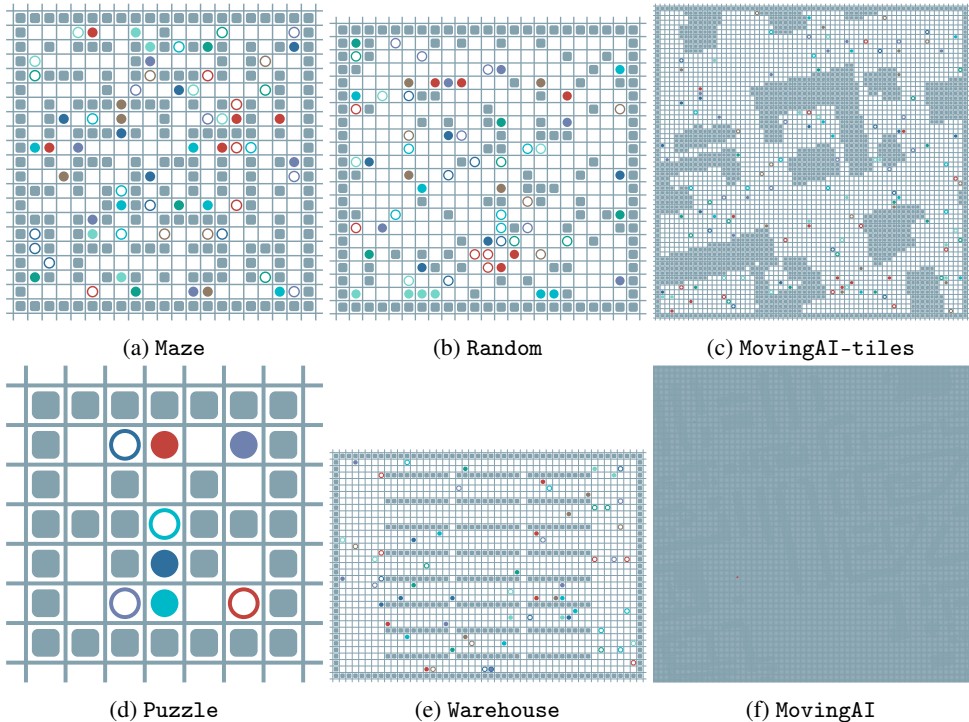

(a) `Maze`  (b) `Random`  (c) `MovingAI-tiles`

(d) `Puzzle`  (e) `Warehouse`  (f) `MovingAI`

Figure 6: Examples of maps presented in POGEMA.

# 5 MARL training setup

For training MARL approaches, such as MAMBA, QMIX, QPLEX, and VDN, we used the default hyperparameters provided in the corresponding repositories[3], and employed the PyMARL2 framework[4] to establish MARL baselines. As input, we apply preprocessing from the Follower approach, which is the current state-of-the-art for decentralized LifeLong MAPF. We attempted to add a ResNet encoder, as used in the Follower approach; however, this addition worsened the results, thus we opted for vectorized observation and default MLP architectures. For centralized methods that work

---

[3]https://github.com/jbr-ai-labs/mamba
[4]https://github.com/hijkzzz/pymarl2

with the state of the environment (e.g., QMIX or QPLEX), we utilized the MARL integration of POGEMA, which provides agent positions, targets, and obstacle positions in a format similar to the SMAC environment (providing their coordinates).

Our initial experiments on training this approach with a large number of agents, similar to the Follower model, showed very low results. We adjusted the training maps to be approximately $16 \times 16$, which proved to be more effective and populated them with 8 agents. This setup shows better results. We continued training the approaches until they reached a plateau, which for most algorithms is under 1 million steps.

## 6    Resources and Statistics

To evaluate all the presented approaches integrated with POGEMA we have used two workstations with equal configuration, that includes 2 NVidia Titan V GPU, AMD Ryzen Threadripper 3970X CPU and 256 GB RAM. The required computation time is heavily depends on the approach by itself.

Table 9: Total time (in hours) required by each of the algorithms to run all MAPF instances on the corresponding datasets.

|  | Random | Mazes | Warehouse | MovingAI-tiles | Puzzles | MovingAI |
|---|---|---|---|---|---|---|
| DCC | 2.11 | 2.46 | 11.07 | 22.70 | 0.09 | 0.02 |
| IQL | 0.05 | 0.04 | 0.13 | 0.13 | 0.01 | 0.01 |
| LaCAM | 0.20 | 0.29 | 0.24 | 0.23 | 0.37 | 0.01 |
| MAMBA | 6.62 | 6.47 | 8.36 | 12.27 | 2.59 | 3.40 |
| QMIX | 0.04 | 0.04 | 0.14 | 0.13 | 0.01 | 0.01 |
| QPLEX | 0.05 | 0.04 | 0.13 | 0.13 | 0.01 | 0.01 |
| SCRIMP | 1.66 | 2.20 | 16.54 | 21.63 | 0.08 | 0.21 |
| VDN | 0.05 | 0.04 | 0.13 | 0.13 | 0.01 | 0.01 |

Table 10: Total time (in hours) required by each of the algorithms to run all LMAPF instances on the corresponding datasets.

|  | Random | Mazes | Warehouse | MovingAI-tiles | Puzzles | MovingAI |
|---|---|---|---|---|---|---|
| ASwitcher | 1.03 | 0.47 | 2.95 | 1.76 | 0.31 | 0.04 |
| EPOM | 0.57 | 0.28 | 0.97 | 0.77 | 0.31 | 0.09 |
| Follower | 0.48 | 0.23 | 0.69 | 0.77 | 0.26 | 0.89 |
| HSwitcher | 6.39 | 2.65 | 18.40 | 10.25 | 0.31 | 0.10 |
| IQL | 0.08 | 0.04 | 0.26 | 0.24 | 0.02 | 0.01 |
| LSwitcher | 6.18 | 2.61 | 17.30 | 10.70 | 0.81 | 0.21 |
| MAMBA | 13.82 | 6.69 | 15.81 | 11.07 | 7.83 | 3.40 |
| MATS-LP | 77.31 | 35.34 | 163.68 | 129.78 | 3.80 | 0.14 |
| QMIX | 0.08 | 0.04 | 0.26 | 0.25 | 0.02 | 0.01 |
| QPLEX | 0.08 | 0.04 | 0.26 | 0.25 | 0.02 | 0.01 |
| RHCR | 0.57 | 0.25 | 17.04 | 6.28 | 0.01 | 0.01 |
| RePlan | 6.00 | 2.40 | 16.20 | 11.33 | 0.01 | 0.09 |
| VDN | 0.08 | 0.04 | 0.25 | 0.25 | 0.02 | 0.01 |

The statistics regarding the spent time on solving MAPF and LMAPF instances are presented in Table 9 and Table 10 respectively. Please note, that all these approaches were run in parallel in multiple threads utilizing dask, that significantly reduces the factual spent time.

We used pretrained models for all the hybrid methods, such as Follower, Switcher, MATS-LP, SCRIMP, and DCC, thus, no resources were spent on their training. RHCR and LaCAM are pure search-based planners and do not require any training. MARL methods, such as MAMBA, QPLEX, QMIX, IQL, and VDN, were trained by us. MAMBA was trained for 20 hours on the MAPF instances, resulting in 200K environment steps, and for 6 days on LifeLong MAPF instances, resulting in 50K environment steps, which corresponds to the same amount of GPU hours. For MARL approaches,

we trained them for 1 million environment steps, which corresponds to an average of 5 GPU hours for each algorithm.

# 7   Accountability framework

Our team is committed to maintaining an open and accountable POGEMA framework. Since 2021, we have continuously improved POGEMA, including the addition of the POGEMA Toolbox and the recent introduction of POGEMA Benchmark. We ensure transparency in our operations and encourage the broader AI community to participate. Our framework includes a fast learning environment, problem instance generator, visualization toolkit, and automated benchmarking tools, all guided by a clear evaluation protocol. We have also implemented/integrated and evaluated multiple strong baselines that simplify further comparison with them. We practice rigorous software testing and conduct regular code reviews. We are promptly addressing issues that are reported on Github and we welcome any feedback and contributions through GitHub.