# OpenReview forum: "POGEMA: A Benchmark Platform for Cooperative Multi-Agent Navigation"
_NeurIPS.cc/2024/Datasets_and_Benchmarks_Track — Submitted to NeurIPS 2024 Track Datasets and Benchmarks_

### Official Review · Reviewer_5xNj · 2024-07-07

**Rating:** 6
**Confidence:** 3

**Review:**

Originality: Multi-Agent Pathfinding (MAPF) is a special type of multi-agent problem, and RL methods have been explored in this research field since the work of PRIMAL [1]. As far as I know, there is a lack of a standard training environment for MAPF. Different papers have developed their own environments, and not all of them are open-sourced, which makes it difficult to compare different methods. Thus, POGEMA can bridge this gap by providing a standard training environment. Currently, six types of environments are provided, such as maze, random, and warehouse, all of which are common test cases in MAPF.

Significance: First, it is significant for the MAPF research field to have a standard RL training environment. Second, it is also helpful for the MARL community to address routing problems. Usually, fully cooperative MARL algorithms are evaluated in domains like SMAC. Routing problems are rarely discussed. As shown by the evaluation results, well-known MARL methods like VDN and QMIX did not perform well in POGEMA.

Concern: My concern on the paper is the training process of the RL methods evaluated in the paper (see Correctness Box)

[1] PRIMAL: Pathfinding via Reinforcement and Imitation Multi-Agent Learning

**Strengths:**

A MAPF training environment and evaluation of MARL algorithms. Significance see Review Box.

**Additional Feedback:**

N/A

**Clarity:**

The paper is overall well written with some items unclear to me (see Opportunities For Improvement Box).

**Correctness:**

+ It is commendable that the authors evaluated several well-known MARL methods such as QMIX, QPLEX, and VDN in POGEMA. However, I noticed in Sec. 5 of the Supplementary material that these methods were trained using default parameters provided by another GitHub repository, typically chosen for training SMAC. This suggests that no parameter sweep was conducted. I suspect this may also apply to the training of other methods like SCRIMP and DCC. Parameter settings, such as learning rate, significantly influence the performance of RL algorithms. Therefore, it would be beneficial for the authors to conduct a parameter sweep when benchmarking these algorithms.

+ The default reward structure in POGEMA is overly sparse, which may not be a good choice for training RL algorithms.

**Documentation:**

Documentation is generally clear.

**Opportunities For Improvement:**

1. Based on my understanding of the paper, the number of agents in each type of environment is fixed at particular values (e.g., Table 1 in Supplementary). It would be more helpful if users could set the number of agents.

2. The MovingAI, MovingAI-tiles, Puzzles in Sec. 4.1 are not well explained, although figures for these three types of environments are shown in Figure 6 in the Supplementary. Please provide more explanation to help readers better understand these cases.

3. There are other types of environments provided by the MAPF benchmark suite at https://movingai.com/benchmarks/mapf/index.html. Incorporating them into POGEMA would be helpful.

4. Lack references for LaCAM, SCRIMP and DCC in Sec. 3.3.

**Relation To Prior Work:**

The related works are discussed in Sec 2.

**Summary And Contributions:**

This paper introduces POGEMA, a Partially-Observable Grid Environment for Multiple Agents. This environment focuses on the multi-agent pathfinding task, which differs from commonly used MARL testbeds like SMAC. POGEMA is developed in Python and provides a gym-like interface. Six different types of domains are provided, and several well-known MARL methods are evaluated in POGEMA.

---

> ### Author Rebuttal · Authors · 2024-08-17
>
> We sincerely thank the reviewer for their thoughtful feedback. We appreciate your recognition of POGEMA’s significance in filling the gap for a standard MAPF training environment and its impact on both the MAPF and MARL communities. Below, we provide responses to the correctness concerns (C1-C2) and opportunities for improvement (I1-I4).
>
> C1: In the Learnable MAPF community it is common practice to provide the weights of trained models alongside the code, as seen with DCC, SCRIMP, and the approaches developed by our team: Follower, MATS-LP, and Switcher. Therefore, we used the readily available weights and did not retrain the models from scratch.
>
> It is indeed unfortunate that most approaches in the MARL community are tuned specifically for the SMAC environment. We agree with the reviewer that proper hyperparameter tuning would enhance our experimental findings, and we are committed to conducting it. Since hyperparameter tuning is a time-consuming process, we started with MAPF instances and tuned the hyperparameters of QPLEX and IQL. For this hyperparameter sweep, we used grid search over parameters such as learning rate, batch size, replay buffer size, and neural network parameters like the size of RNN blocks. We used the default functionality of the Wandb framework for this sweep, with the optimization target being the CSR of the agent on the training maps.
>
> The ranges of the values we tested were:
> • Batch size: 16, 32, 64, 128, 256, 512
> • Buffer size: 1000, 2000, 5000, 10000
> • Learning rate: 0.002, 0.001, 0.0005, 0.00025
> • RNN size: 32, 64, 128, 256
>
> The best parameters for the algorithms are:
> • For QPLEX: Batch size 64, buffer size 5000, learning rate 0.002, RNN hidden state size 256.
> • For IQL: Batch size 64, buffer size 5000, learning rate 0.002, RNN hidden state size 64.
>
> We include the webplot and the results on random and maze maps in the PDF file. It can be seen that these MARL algorithms have improved, but they still underperform compared to other approaches.
>
> C2: Indeed, the default reward in the POGEMA environment is sparse, and we agree that this can make training RL approaches challenging. We chose this reward structure to provide a simple default option that is less prone to exploitation. Additionally, we deliberately avoid imposing specific biases in the reward function, allowing researchers the flexibility to explore various approaches. Reward shaping is often part of learnable MAPF methods, so we do not restrict it in our benchmark. However, it’s clear that a strong policy can still be trained with a sparse reward, as shown by the Switcher approach using POGEMA’s default reward.
>
> I1: It’s possible to set any viable number of agents and the POGEMA environment would generate an instance for it. We show the chosen numbers of agents in Table 1 for reproducibility of the results. It also allows other researchers to evaluate their own approaches on the same instances with the same number of agents and reuse the provided results without need to reconduct the experimental evaluation of the baselines.
>
> I2: We will describe the dataset and available type of maps more clearly in the main text. Due to the lack of space, some details of explanation of the dataset were omitted. From the original MovingAI benchmark we have picked city maps with 256x256 size. Due to the large size of the maps we have decided to create and additional dataset (MovingAI-tiles), where we have sliced the original maps on 16 tiles and received maps of 64x64 size. Puzzles dataset consists of 16 tiny (5x5) maps. The puzzles maps mainly consist of narrow 1-cell corridors (like mazes), but in contrast to mazes, they almost have no alternative passages, that require agents to perform complex cooperative actions to resolve collisions and surpass each other.
>
> I3: We are going to integrate a script into POGEMA that would be able to convert the maps/scenarios presented in MovingAI benchmark to the format compatible with POGEMA. We are not able to directly integrate all data from MovingAI benchmark into POGEMA due to potential license issues (especially with the maps obtained from computer games).
>
> I4: The references to these approaches are presented in Table 2 with the overview of approaches, but we should definitely add the references to Sec. 3.3. where these approaches are mentioned for the first time.

---

> > ### Author Response · Authors · 2024-08-29
> > **Incorporating MovingAI maps into POGEMA and MARL hyperparameter sweep**
> >
> > ## Incorporating MovingAI maps into POGEMA
> >
> >   Here we incorporated an ingestion script to convert MovingAI maps to be compatible with POGEMA. The ingestion script is straightforward and is implemented in the following link: [MovingAI Ingestion Script](https://github.com/Cognitive-AI-Systems/pogema-toolbox/blob/main/pogema_toolbox/moving_ai_ingestion.py).
> >
> > Here is an example of usage:
> >
> > ```python
> > from pogema_toolbox.generators.generator_utils import maps_dict_to_yaml
> > from pogema_toolbox.moving_ai_ingestion import download_moving_ai_maps
> >
> > url = 'https://movingai.com/benchmarks/street/street-map.zip'
> > maps = download_moving_ai_maps(url)
> > maps_dict_to_yaml('maps.yaml', maps)
> > ```
> >
> > This script downloads the full archive of, in this case, the street-map series, which will be saved to a YAML file compatible with POGEMA.
> >
> >
> >   ## MARL hyperparameter sweep for LifeLong MAPF
> > | Model       | Cooperation | Scalability | Congestion | Out-of-Distribution | Performance | Pathfinding |
> > |-------------|-------------|-------------|------------|---------------------|-------------|-------------|
> > | RHCR        | 0.9870      | 0.2744      | 0.4887     | 0.9744              | 0.9994      | 1.0000      |
> > | IQL         | 0.0814      | 1.0         | 0.4710     | 0.1734              | 0.1454      | 0.0375      |
> > | IQL-tuned   | 0.2377      | 1.0         | 0.9421     | 0.5704              | 0.4543      | 0.1125      |
> > | QPLEX       | 0.1019      | 1.0         | 0.6944     | 0.4269              | 0.3786      | 0.1125      |
> > | QPLEX-tuned | 0.4025      | 1.0         | 0.7967     | 0.8498              | 0.6538      | 0.7875      |
> > | QMIX        | 0.2150      | 1.0         | 0.8583     | 0.4821              | 0.2429      | 0.1250      |
> > | QMIX-tuned  | 0.4147      | 1.0         | 0.8922     | 0.8024              | 0.6427      | 0.2625      |
> > | VDN         | 0.0590      | 1.0         | 0.7344     | 0.2530              | 0.1619      | 0.0375      |
> > | VDN-tuned   | 0.2523      | 1.0         | 0.7223     | 0.6534              | 0.4572      | 0.0500      |
> >
> > Here we present the results for previous metrics of MARL algorithms and their tuned versions. The best hyperparameters found are as follows:
> >
> > The best found hyperparameters:
> > • IQL: batch_size = 64, buffer_size = 5000, lr = 0.002, rnn_hidden_dim = 128
> > • QPLEX: batch_size = 64, buffer_size = 5000, lr = 0.002, rnn_hidden_dim = 128
> > • QMIX: batch_size = 32, buffer_size = 5000, lr = 0.001, rnn_hidden_dim = 256
> > • VDN: batch_size = 32, buffer_size = 5000, lr = 0.001, rnn_hidden_dim = 256
> >
> > It can be seen that the results significantly improved after tuning for all algorithms on Performance, Out-of-Distribution, Cooperation maps and Pathfinding.
> >
> > ## MARL hyperparameter sweep for MAPF
> >
> >
> > | Model       | Cooperation | Scalability | Congestion | Out-of-Distribution | Performance | Pathfinding |
> > |-------------|-------------|-------------|------------|---------------------|-------------|-------------|
> > | IQL         | 0.1362      | 1.0         | 0.5027     | 0.0                 | 0.0         | 0.0375      |
> > | IQL-tuned   | 0.4014      | 1.0         | 0.7472     | 0.0                 | 0.1242      | 0.075      |
> > | QPLEX       | 0.3704      | 1.0         | 0.7348     | 0.0                 | 0.0860      | 0.1125      |
> > | QPLEX-tuned | 0.4729      | 1.0         | 0.8107     | 0.0                | 0.6518      | 0.0625      |
> > | QMIX        | 0.4359      | 1.0         | 0.8415     | 0.0                 | 0.0116      | 0.125      |
> > | QMIX-tuned  | 0.3261      | 1.0         | 0.8531     | 0.0                 | 0.2810      | 0.1      |
> > | VDN         | 0.0971      | 1.0         | 0.7989     | 0.0                 | 0.0         | 0.0375      |
> > | VDN-tuned   | 0.4370      | 0.9379      | 0.7696     | 0.0                 | 0.0548      | 0.025      |
> >
> > Here we report the metrics of algorithms after a hyperparameter sweep in comparison to previous versions. We have excluded the results of the centralized approach (LaCAM) to better highlight the difference in scores between the tuned and untuned versions. When comparing MARL approaches to other algorithms, they are still underperforming.
> >
> > The best found hyperparameters:
> >
> > • IQL: batch_size = 64, buffer_size = 5000, lr = 0.002, rnn_hidden_dim = 64
> > • QPLEX: batch_size = 64, buffer_size = 5000, lr = 0.002, rnn_hidden_dim = 256
> > • QMIX: batch_size = 32, buffer_size = 2000, lr = 0.002, rnn_hidden_dim = 64
> > • VDN: batch_size = 256, buffer_size = 5000, lr = 0.001, rnn_hidden_dim = 64
> >
> >
> > Please let us know if any additional clarifications are needed.

---

> > > ### Comment · Reviewer_5xNj · 2024-08-30
> > >
> > > Thanks authors for the detailed explanation and additional results on parameter searching and new maps! So please incorporate them in the revised version. Most of my concerns are addressed except for the sparse reward. I partially agree with the author that sparse reward is like providing a default setting. But if we check other RL MAPF papers, e.g., PRIML, DHC, DCC, they all designed some dense rewards. So I still encourage the authors to try some dense reward design. But I am happy to support the acceptance of the paper given current materials.

---

> > ### Author Response · Authors · 2024-08-30
> > **Official Comment**
> >
> > We agree with the reviewer regarding the issue of sparsity versus density in reward functions. However, in the case of MARL algorithms, for example, we used dense rewards from the Follower approach, rewarding the agent with $+r$ for reaching waypoints (as described in the paper). As mentioned, we are leaving sparse rewards as a default option only.
> >
> > The question you raised is indeed important, and as a potential solution, we propose adding a list of wrappers that would implement the reward functions of popular learnable MAPF algorithms such as PRIMAL, DCC, and DHC, etc. We are committed to implementing and providing these reward functions.

---

### Official Review · Reviewer_dFGp · 2024-07-19
**Useful multi-agent navigation tool suite**

**Rating:** 7
**Confidence:** 3
**Clarity:** Yes.

**Review:**

- Quality: the authors present a comprehensive set of tools for the multi-agent navigation problem, including a training environment, state-of-the-art baselines, multiple categories of maps, an evaluation protocol with multiple metrics, and visualization tools. These tools, which make up the POGEMA suite, are customizable and extensible, making them useful to the multi-agent navigation and MARL communities.
- Clarity: the contributions and the environment are presented clearly and effectively.
- Originality: while other multi-agent navigation environments and benchmarks exist, POGEMA includes some additional features (e.g., native Python, hardware-agnostic, procedural map generation) that allow for more comprehensive evaluation and ease of use by researchers.

**Strengths:**

Various useful features of POGEMA:

- Straightforward API with opportunity for customization and extensions
- Custom maps
- Fast, distributed evaluation using Dask
- Visualization tools for analysis
- Map generation tool. Can make random maps, mazes, and warehouse maps
- Integrates SOTA baselines for planning-based, hybrid, and learning-based multi-agent planners.
- Evaluation protocol with multiple metrics

Additional strengths of the paper:

- Evaluate SOTA algorithms that use a variety of methods (planning-based, learning-based, hybrid) and showcase the evaluation protocol by presenting metrics for these algorithms.
- Highlight the poor performance of MARL algorithms compared to centralized planners and and other specialized, learning-based algorithms.

**Additional Feedback:**

- There are minor grammatical mistakes throughout the paper, particularly related to the usage or omission of definite and indefinite articles (e.g., “the”, “a”, “an”).
- Line 138: “environemnt” —> “environment”
- Line 217: “ontains” —> “contains”
- Line 239: “it’s” —> “it”
- Figure 3: the caption mentions “(a)” and “(b)”, which aren’t actually labeled in the figure, though it is easy to assume that they refer to the left and right plots, respectively. It would be more clear to change the caption to have “(left)” and “(right)” or to add the (a) and (b) sub-captions to the plots.

**Correctness:**

The main correctness issue is regarding the analysis of the LMAPF results (see Opportunities For Improvement comments).

As a benchmark, the evaluation methods are generally sound, but some improvements could be made (also mentioned in Opportunities For Improvement):
- Measure collisions as a metric and possibly include this in the reward or end the episode (or freeze the agent) if a collision occurs.
- Some metrics are not described very clearly or should be renamed.

**Documentation:**

The documentation and usage are presented clearly, and the authors maintain a GitHub repo for the benchmark, which they maintain.

**Ethics:**

No ethical concerns.

**Limitations:**

The authors briefly mention the limitations in section 5, but it would be helpful to get a more in-depth explanation (at least in the appendix), as well as plans for addressing the limitations.

The authors did not address the potential negative societal impact of their work.

**Opportunities For Improvement:**

- The fact that POGEMA is purely Python-based makes me expect it to be slower than other simulators that use hardware acceleration with GPUs and TPUs. It would be useful to see the compute resources and runtimes used by POGEMA compared to other simulators for a fair comparison.
- It is unclear what POGEMA offers as far as training agents/policies for multi-agent navigation. My understanding is that POGEMA offers a MARL integration for training policies, but I assume this training is not purely Python-based and requires ML training libraries like Jax or PyTorch. If this is the case, would these libraries not also be needed for the evaluation of these policies? The claim that POGEMA is purely Python-based is difficult to understand.
- POGEMA prevents collisions automatically (line 143), but collision avoidance is often part of the challenge of robotic navigation tasks, whether single-agent or multi-agent. It would be beneficial for POGEMA to allow for collisions and include collision-related metrics to measure how well different algorithms implement collision avoidance.
- The authors claim that the dimensionality of the action space grows grows exponentially with the number of agents (line 32), but in a decentralized approach where each agent is running its own policy, the action space is the same for MARL as it would be for single-agent RL.
- For the performance metrics (section 4.2):
    - It’s unclear what SoC_best and SoC refer to. Is SoC_best the SoC for the best-performing approach, and SoC is for the current approach being evaluated? This should be made more clear.
    - The “Cooperation” metric seems to be a misnomer since it just tests performance in the Puzzles category. Perhaps this tests complexity, but I do not see why it tests cooperation specifically. If the Puzzles maps are specifically designed to require a high degree of cooperation for success, this should be stated and explained.
    - The pathfinding metric would be more useful if it were a continuous measure of the quality of the path. For example, it could be the ratio of the optimal path to the path found by the approach being evaluated.
- For MAPF, it is surprising that the scalability of centralized planners is as good, if not better, than that of decentralized planners. I would have expected one advantage of decentralized planners is their ability to scale more effectively to very large numbers of agents. It would be useful for the authors to help explain or interpret these findings.
- The authors claim that Follower outperforms RHCR on the Performance metric for LMAPF (lines 286-287), but the right plot in Figure 3 suggests otherwise. RHCR (orange) achieves a higher Performance metric than Follower (blue). They also claim that Follower doesn’t do worse than RHCR significantly on the Cooperation metric, but RHCR gets 100 while Follower gets around 65, which is much worse.

**Relation To Prior Work:**

Yes, the authors show an in-depth feature comparison with many other environments/benchmarks.

**Summary And Contributions:**

- This paper presents POGEMA, a suite of tools for evaluating multi-agent navigation algorithms.
- Key features of POGEMA include a straightforward API; customization of the environment; custom maps in various categories; fast, distributed evaluation using Dask; visualization tools for analysis; a map generation tool; and integrated SOTA baselines for planning-based, hybrid, and learning-based multi-agent planners.
- The authors evaluate multiple state-of-the-art multi-agent navigation methods across different environments and report multiple metrics, highlighting the benefits of centralized planners and the drawbacks of certain MARL algorithms.

---

> ### Author Rebuttal · Authors · 2024-08-17
>
> We sincerely thank the reviewer for their thorough and positive feedback. We appreciate your recognition of POGEMA’s comprehensive set of tools and its contributions to the multi-agent navigation and MARL communities. Your acknowledgment of the strengths, such as the customizable API, map generation tools, and integrated baselines, reinforces our commitment to making POGEMA. Below, we address the opportunities for improvement (I1-I7) highlighted in the review.
>
> I1: We agree that the speed of the environment is a crucial aspect of any RL setup. We kindly ask you to refer to our response to reviewer [BG3N](https://openreview.net/forum?id=DG4k8PJ8qv&noteId=7nZ6A2L8zE), where we provide detailed information on the speed performance. While POGEMA is expected to be slower in very large vectorized setups compared to environments using GPU or TPU hardware acceleration, this trade-off has an advantage. By not relying on GPUs or TPUs for environment simulation, these resources remain fully available for training neural networks, which often represents the primary bottleneck in large-scale RL experiments.
>
> I2: Our claim that POGEMA is purely Python-based refers specifically to the environment itself, not the training setups.  POGEMA does offer MARL integration for training policies. However, this training process is not purely Python-based and does require machine learning libraries such as PyTorch or JAX. These libraries would also be necessary for evaluating the trained policies. We will explicitly mention this in the paper to clarify any doubts.
>
> I3: Thank you for the idea to measure the number of prevented collisions and utilize it as an additional metric. We will definitely perform an additional series of experiments to evaluate this metric and provide you the results in the next few days. It is also worth noting that POGEMA has multiple collision systems that differ in the way the collisions are prevented. For example, if the ‘block_both’ system is chosen, then none of the agents is moved, if two or more agents try to move to the same cell simultaneously. Another system, called ‘soft’, allows one of the agents to move to this cell in the same situation. It’s possible to integrate another collision system that would not only block the current action, but also freeze the agent for the next step (or few of them, or even until the end of the episode).
>
> I4: We agree with the reviewer in that in decentralized MARL there is no exponential growth of the action space. Our phrase implicitly assumed the centralized MARL, i.e. when a single decision-making policy picks a joint action which is an element of a cardinal product of the individual actions sets.
>
> I5.1: We agree with the reviewer and will explain this point in the text more clearly. You are right, to compute some of the metrics, such as performance, we use the ratio between SoC (obtained by current approach being evaluated) and SoC_best - the lowest (best) SoC value for a particular instance received by any of the evaluated approaches. In our experiments, the best SoC value most of the time was obtained by LaCAM, as it’s a classical centralized approach. However, there were still some rare cases where the SoC_best value was obtained by other approaches (DCC/SCRIMP). In the ideal world instead of SoC_best we would use SoC_optimal. Unfortunately, it’s hard to obtain them, especially for the dense scenarios and the ones that contain hundreds of agents. Alternatively, we can use the lower bound of SoC_optimal value, i.e. sum of the costs of the individual shortest paths. However, this would prevent any approach from achieving a 100% metric, which we want to avoid.
>
> I5.2: We agree with the reviewer. The puzzles maps were specifically designed by us to contain narrow corridors, deadlocks and lacunas. These problem instances require agents to use specific layout patterns to cooperate, such as one agent entering a corridor and using a lacuna to allow another agent to pass. We will emphasize this more in the revised manuscript.
>
> I5.3: We have modified the measure of the pathfinding metric according to your suggestion. Now it’s equal to the ratio between the cost of the optimal path and the found one. There might also be a case where no path is found. In that case, the metric is zero.
>
> I6: You are right, and scalability of centralized approaches in most of the cases suffers with the increasing complexity of the task that is usually controlled by the number/density of the agents. For the MAPF scenarios as a centralized classical approach we have utilized the LaCAM approach, which is different in the way of obtaining solutions from such centralized planners as CBS, PBS, BCP, etc. which are tailored to find good (or even optimal) solutions from scratch. Instead, LaCAM first finds a solution of low quality utilizing a rule-based approach (PIBT). Having an allocated time budget it tries to improve the quality of the solution. As a result, it has a great scalability, however, the quality of its solutions might degrade significantly on the instances with very large numbers of agents.
>
> It’s also worth noting the results of another centralized approach presented in the paper—RHCR. While it incorporates some time-budget techniques, it relies on the PBS approach, which struggles with scalability to a large number of agents. The poor scalability of the RHCR approach is evident in the results shown in the paper.
>
> The results presented in the paper also demonstrate the poor scalability of certain learning-based approaches, such as SCRIMP and DCC. This behaviour is due to the presence of communication and tie-breaking mechanisms (in SCRIMP), which prevent these approaches from scaling linearly as the number of agents increases.
>
> I7: Thank you for pointing out these incorrect claims. The explanation of the experiments will certainly be revised. The claims were based on previous results where Follower outperformed RHCR due to incorrect evaluation parameters of RHCR.

---

> > ### Author Response · Authors · 2024-08-27
> > **Collision metric**
> >
> > We have re-evaluated most approaches on Mazes and Random map sets to measure the collision metric. We identified two types of collisions: between agents and with obstacles. Collisions are recorded when an agent remains in the same position while performing a non-waiting action.
> >
> > There are no results for LaCAM, as it does not encounter collisions. SCRIMP also has no results since it uses its own environment for communication and tie-breaking, ensuring collision-free actions. It's also worth noting that collisions with obstacles are easily avoided by masking actions that lead to them.
> >
> > The averaged results on MAPF instances are shown below, with the best values highlighted in bold. In all cases, DCC achieved the fewest collisions.
> >
> > | Number of Agents | Algorithm | Random       | Random       | Mazes        | Mazes        |
> > |------------------|-----------|--------------|--------------|--------------|--------------|
> > |                  |           | Agents Collisions | Obstacle Collisions | Agents Collisions | Obstacle Collisions |
> > | 8                | DCC       | **1.59**     | **0.00**     | **1.77**     | **0.15**     |
> > | 8                | IQL       | 2.45         | 54.09        | 3.83         | 57.61        |
> > | 8                | MAMBA     | 5.50         | 8.38         | 13.77        | 3.02         |
> > | 8                | QMIX      | 2.93         | 48.61        | 3.67         | 46.56        |
> > | 8                | QPLEX     | 3.08         | 13.69        | 3.75         | 12.15        |
> > | 8                | VDN       | 3.06         | 114.28       | 5.37         | 17.12        |
> > |------------------|-----------|--------------|--------------|--------------|--------------|
> > | 16               | DCC       | **7.26**     | **0.02**     | **8.64**     | **0.27**     |
> > | 16               | IQL       | 12.99        | 140.63       | 23.55        | 100.05       |
> > | 16               | MAMBA     | 39.04        | 29.75        | 64.34        | 17.30        |
> > | 16               | QMIX      | 10.65        | 99.84        | 17.04        | 91.73        |
> > | 16               | QPLEX     | 18.38        | 38.24        | 21.84        | 31.92        |
> > | 16               | VDN       | 24.34        | 293.96       | 26.76        | 62.23        |
> > |------------------|-----------|--------------|--------------|--------------|--------------|
> > | 24               | DCC       | **16.99**    | **0.11**     | **22.20**    | **0.55**     |
> > | 24               | IQL       | 33.03        | 223.80       | 64.70        | 168.88       |
> > | 24               | MAMBA     | 118.66       | 274.31       | 193.40       | 195.25       |
> > | 24               | QMIX      | 35.93        | 189.05       | 57.94        | 151.95       |
> > | 24               | QPLEX     | 53.04        | 81.12        | 70.87        | 67.06        |
> > | 24               | VDN       | 83.63        | 560.53       | 85.17        | 149.65       |
> > |------------------|-----------|--------------|--------------|--------------|--------------|
> > | 32               | DCC       | **38.76**    | **0.14**     | **49.62**    | **0.87**     |
> > | 32               | IQL       | 85.52        | 319.73       | 140.55       | 287.98       |
> > | 32               | MAMBA     | 298.47       | 814.38       | 427.45       | 662.20       |
> > | 32               | QMIX      | 82.74        | 339.23       | 139.34       | 275.25       |
> > | 32               | QPLEX     | 132.96       | 161.88       | 169.83       | 124.03       |
> > | 32               | VDN       | 217.95       | 946.48       | 228.28       | 299.49       |
> > |------------------|-----------|--------------|--------------|--------------|--------------|
> > | 48               | DCC       | **125.88**   | **0.54**     | **141.45**   | **2.76**     |
> > | 48               | IQL       | 470.90       | 838.77       | 469.88       | 1047.49      |
> > | 48               | MAMBA     | 1147.41      | 1477.06      | 1260.41      | 1455.84      |
> > | 48               | QMIX      | 316.77       | 897.48       | 533.98       | 761.66       |
> > | 48               | QPLEX     | 564.48       | 551.81       | 652.95       | 412.75       |
> > | 48               | VDN       | 965.25       | 1942.29      | 939.67       | 845.27       |
> > |------------------|-----------|--------------|--------------|--------------|--------------|
> > | 64               | DCC       | **310.20**   | **1.16**     | **285.27**   | **2.77**     |
> > | 64               | IQL       | 1494.66      | 1660.95      | 1081.20      | 2145.54      |
> > | 64               | MAMBA     | 2618.98      | 2138.71      | 2505.97      | 2198.84      |
> > | 64               | QMIX      | 909.52       | 1767.12      | 1270.92      | 1518.47      |
> > | 64               | QPLEX     | 1961.10      | 1486.55      | 1789.67      | 1066.40      |
> > | 64               | VDN       | 2382.54      | 3029.23      | 2404.80      | 1715.71      |
> >
> > The results for LMAPF instances are given in the next comment due to the characters limit in one message.

---

> > > ### Author Response · Authors · 2024-08-27
> > > **Collision metric LMAPF**
> > >
> > > The averaged results on LMAPF instances are shown below, with the best values highlighted in bold.
> > > MATS-LP has no obstacle collisions as it utilizes masking.
> > >
> > > |Agents|Algorithm | Random       | Random      | Mazes        | Mazes       |
> > > |------|----------|--------------|-------------|--------------|-------------|
> > > |      |          | Agents Coll. | Obst. Coll. | Agents Coll. | Obst. Coll. |
> > > |8     |ASwitcher | 21.91        | 12.57       | 7.45         | 4.38        |
> > > |8     |Follower  | 42.39        | **2.94**    | 31.98        | **1.49**    |
> > > |8     |IQL       | **3.73**     | 139.32      | 8.86         | 43.77       |
> > > |8     |LSwitcher | 19.66        | 16.17       | **6.76**     | 5.79        |
> > > |8     |MAMBA     | 200.15       | 11.41       | 137.06       | 4.48        |
> > > |8     |MATS-LP   | 23.78        | 0.00        | 19.69        | 0.00        |
> > > |8     |QMIX      | 14.26        | 246.08      | 7.78         | 25.31       |
> > > |8     |QPLEX     | 36.74        | 37.95       | 14.70        | 13.84       |
> > > |8     |VDN       | 9.55         | 264.33      | 13.45        | 26.48       |
> > > |------|----------|--------------|-------------|--------------|-------------|
> > > |16    |ASwitcher | 98.37        | 44.55       | 36.18        | 21.24       |
> > > |16    |Follower  | 187.89       | **11.59**   | 146.03       | **4.81**    |
> > > |16    |IQL       | **30.41**    | 322.35      | 43.53        | 73.97       |
> > > |16    |LSwitcher | 90.85        | 51.16       | **36.03**    | 21.50       |
> > > |16    |MAMBA     | 799.27       | 38.86       | 506.44       | 15.18       |
> > > |16    |MATS-LP   | 114.85       | 0.00        | 93.27        | 0.00        |
> > > |16    |QMIX      | 86.75        | 555.56      | 43.02        | 54.11       |
> > > |16    |QPLEX     | 167.31       | 114.15      | 66.84        | 26.93       |
> > > |16    |VDN       | 68.97        | 648.73      | 64.83        | 63.43       |
> > > |------|----------|--------------|-------------|--------------|-------------|
> > > |24    |ASwitcher | 243.01       | 112.88      | **87.30**    | 53.28       |
> > > |24    |Follower  | 429.45       | **29.26**   | 353.02       | **9.33**    |
> > > |24    |IQL       | **99.59**    | 512.52      | 111.92       | 130.66      |
> > > |24    |LSwitcher | 230.62       | 124.90      | 90.30        | 50.32       |
> > > |24    |MAMBA     | 1836.36      | 70.33       | 1023.59      | 28.42       |
> > > |24    |MATS-LP   | 281.41       | 0.00        | 250.62       | 0.00        |
> > > |24    |QMIX      | 249.91       | 977.83      | 112.83       | 109.59      |
> > > |24    |QPLEX     | 391.27       | 219.18      | 174.02       | 62.21       |
> > > |24    |VDN       | 216.38       | 1181.03     | 145.20       | 136.05      |
> > > |------|----------|--------------|-------------|--------------|-------------|
> > > |32    |ASwitcher | 453.00       | 222.95      | **166.05**   | 106.97      |
> > > |32    |Follower  | 745.77       | **49.00**   | 635.09       | **13.38**   |
> > > |32    |IQL       | **245.62**   | 771.66      | 198.38       | 277.38      |
> > > |32    |LSwitcher | 422.09       | 224.86      | 172.77       | 93.91       |
> > > |32    |MAMBA     | 2932.12      | 115.89      | 1630.87      | 42.18       |
> > > |32    |MATS-LP   | 558.62       | 0.00        | 576.41       | 0.00        |
> > > |32    |QMIX      | 533.95       | 1520.92     | 258.99       | 215.25      |
> > > |32    |QPLEX     | 725.46       | 451.35      | 337.55       | 127.08      |
> > > |32    |VDN       | 527.48       | 1913.23     | 306.73       | 206.56      |
> > > |------|----------|--------------|-------------|--------------|-------------|
> > > |48    |ASwitcher | 1157.16      | 603.83      | **438.27**   | 312.82      |
> > > |48    |Follower  | 1607.15      | **116.28**  | 1418.89      | **23.86**   |
> > > |48    |IQL       | 1781.20      | 2293.56     | 606.89       | 1119.61     |
> > > |48    |LSwitcher | **1046.27**  | 501.68      | 456.91       | 243.98      |
> > > |48    |MAMBA     | 6067.58      | 170.88      | 3153.30      | 58.09       |
> > > |48    |MATS-LP   | 1540.98      | 0.00        | 1867.37      | 0.00        |
> > > |48    |QMIX      | 1679.02      | 2966.17     | 788.35       | 682.90      |
> > > |48    |QPLEX     | 2319.12      | 1523.77     | 927.22       | 439.84      |
> > > |48    |VDN       | 1974.20      | 3937.16     | 953.18       | 781.44      |
> > > |------|----------|--------------|-------------|--------------|-------------|
> > > |64    |ASwitcher | 2308.62      | 1200.01     | **897.02**   | 696.16      |
> > > |64    |Follower  | 2700.09      | 231.75      | 2473.05      | **43.47**   |
> > > |64    |IQL       | 4351.40      | 3919.01     | 1165.72      | 2155.73     |
> > > |64    |LSwitcher | **1983.53**  | 857.52      | 927.49       | 490.77      |
> > > |64    |MAMBA     | 10054.48     | **225.09**  | 4898.24      | 78.23       |
> > > |64    |MATS-LP   | 3354.27      | 0.00        | 4356.56      | 0.00        |
> > > |64    |QMIX      | 3372.20      | 4669.86     | 1599.48      | 1344.98     |
> > > |64    |QPLEX     | 5882.98      | 3498.84     | 2115.50      | 1103.86     |
> > > |64    |VDN       | 4830.05      | 6104.47     | 2253.84      | 1617.23     |

---

### Official Review · Reviewer_BG3N · 2024-07-22
**Review of POGEMA**

**Rating:** 6
**Confidence:** 3
**Correctness:** Yes
**Clarity:** Good

**Review:**

I feel like the environment is simpler than non-grid based environment that existing. If the proposed env has advantages of speed, it is useful considering the small observation and action space.

**Strengths:**

The environment is useful for multiple agent RL due to its simplicity

**Additional Feedback:**

My main question is the speed and parallelization capacity of the environment which is often the bottlenecks.

**Documentation:**

Good

**Opportunities For Improvement:**

Please provide more information on the speed of the env which are often the bottlenecks

**Relation To Prior Work:**

Good

**Summary And Contributions:**

The paper provides a simulator for multiple-agent navigation. The map is grid-based which reduces the action space compared to the existing/well-used multiple-agent environment.  My main question is the speed and parallelization capacity of the environment which is often the bottlenecks.

---

> ### Author Rebuttal · Authors · 2024-08-17
>
> We thank the reviewer for their comments and appreciate the acknowledgment of POGEMA’s usefulness for multi-agent RL. Below, we provide a detailed response to the question regarding speed performance.
>
> The speed of the environment is a crucial aspect of any RL environment, often playing a key role in training performance and usability. First of all, we provide brief information about the speed of the multi-agent environments in Table 1, which shows that the environment can handle more than 10K steps per second. However, we agree that providing more in-depth information about the POGEMA is an important point.
>
> Continuous integration (CI) for POGEMA already includes a speed performance measurement procedure, which runs alongside tests on each push to the main branch of the repository. Here, we provide the results of the latest CI run: [Link to CI Results](https://github.com/AIRI-Institute/pogema/actions/runs/10055580665/job/27792532086). Please note that these results were obtained using the default GitHub instance for Actions; performance on a faster server-side CPU would likely be better. Nevertheless, this test provides a good measure of performance on an average setup.
>
> For this test, we used the default observation parameters commonly employed in learnable MAPF approaches (such as Follower, SCRIMP, DCC, etc.). Specifically, we set obs_radius = 5, which corresponds to an 11x11 field. We provide the results for both MAPF and LMAPF scenarios with random policy.
> For scenarios with more than 32 agents, POGEMA achieves SPS ≥ 70k, which is notably fast for a single-environment and single-thread setup. For comparison, we can reference [EnvPool](https://github.com/sail-sg/envpool), which provides 50k FPS for Atari on a 12-CPU setup. It’s important to note that the reported SPS refers to the total frames (observations) received by all agents, not the environment steps.
>
> | num_agents | size | MAPF (SPS) | LMAPF (SPS) |
> |------------|------|---------------|---------------|
> |       1 |   32 | 14236.2  |     17862.3  |
> |       1 |   64 |     9228.35 |      8670.55 |
> |       32 |   32 |    87197.1  |     79638.9  |
> |       32 |   64 |    76115  |      74830.5  |
> |       64 |   32 |    91364.8  |     90254.8  |
> |       64 |   64 |    84719.6  |     83455.1  |
>
> The speed of POGEMA can be further improved by running more asynchronous environments in parallel. We can refer to the Follower approach ([Follower](https://github.com/AIRI-Institute/learn-to-follow)), which successfully utilizes the modern SampleFactory framework ([SampleFactory](https://github.com/alex-petrenko/sample-factory)) for large-scale RL training. POGEMA has built-in integration with this framework. In our experience, the primary bottleneck is GPU computational power. In the Follower paper, a fairly large RL network with a ResNet encoder and GRU heads was trained in 18 hours for 1 billion steps. In conclusion, POGEMA is an efficient, scalable environment for multi-agent RL, especially with asynchronous execution and integration with frameworks like SampleFactory.

---

### Author Rebuttal · Authors · 2024-08-19

We sincerely thank the reviewers for their time and expertise. The insightful feedback provided will be instrumental in refining our paper. We appreciate the recognition from the reviewers that the paper introduces several useful features in POGEMA, such as a straightforward API with opportunities for customization (Reviewer dFGp), a robust map generation tool, and fast, distributed evaluation using Dask (Reviewer dFGp). Additionally, we are grateful for the acknowledgment of the paper's significant contribution to the MAPF and MARL research fields by providing a standard RL training environment and evaluating state-of-the-art algorithms across various metrics (Reviewers dFGp and 5xNj). We also value the positive comments on the clarity of the paper (Reviewers dFGp, BG3N, and 5xNj) and the thorough documentation and usage presented (Reviewers dFGp and BG3N).

During the review phase, we published a preprint version of the POGEMA Benchmark paper on arXiv. We have not included the link here, as we do not have the right to ask the reviewers to look at the updated version of the manuscript. However, we would like to highlight a list of additions and changes that address some of the questions raised and support our initial claims.

1. We further enhanced the Related Work section by adding descriptions for each column of Table 1. For instance, the description of the column "Performance >10K Steps/s" partly addresses the question about speed performance. Here is the description:

Performance >10K Steps/s Training and evaluating multi-agent reinforcement learning agents
often requires making billions of steps (transitions) in the environment. Thus, it is crucial that each
transition is computed efficiently. In general, performing more than 10K steps per second is a good
indicator of the environment’s efficiency. While XLA versions can provide high performance by
vectorizing the environment on GPU, they require modern hardware setups, which can be a barrier
for some researchers. In contrast, fast environments like POGEMA or RWARE can achieve high
performance without such stringent hardware requirements, making them more accessible and easier
to integrate into a variety of research projects.

We are committed to further enhancing this by providing an additional section on POGEMA's speed performance, based on our response to reviewer BG3N.

2. We have published all the packages on GitHub under permissive licences, and we also support PyPI packages for POGEMA and the POGEMA toolbox. Here is the full list of GitHub repositories: [pogema](https://github.com/AIRI-Institute/pogema), [pogema-benchmark](https://github.com/AIRI-Institute/pogema-benchmark), [pogema-toolbox](https://github.com/AIRI-Institute/pogema-toolbox).

3. We improved the visualization of the Experimental Results section and corrected the mistaken claim regarding the performance of Follower and RHCR, as highlighted by reviewer dFGp. Furthermore, we enhanced the visualization for the in-depth results description in the appendix, highlighting both the best-performing solution and the best-performing learnable solution in the tables.

4. We have provided an Extended Related Work section in the appendix, which thoroughly describes each environment listed in Table 1.


In response to the reviewers, we have agreed to provide additional improvements, which we planned to implement during the discussion period. Here is the list of them:
- A hyperparameter sweep for MARL approaches, as requested by reviewer 5xNj.
- A MovingAI maps ingestion script, which we planned to include in the POGEMA toolbox, also in response to a request from reviewer 5xNj.
- An additional series of experiments to evaluate the collision metric, as requested by reviewer dFGp.

---

### Decision · Program_Chairs · 2024-09-26

**Decision:**

Reject

**Comment:**

This paper presents POGEMA, an environment for multi-agent pathfinding (MAPF) and multi-agent reinforcement learning (MARL), featuring a customizable API, map generation, and distributed evaluation. Reviewers acknowledged its utility but raised concerns about the environment's necessity, speed performance, and experimental depth. During the discussion, the authors addressed these points by enhancing the Related Work section, clarifying performance metrics, correcting experimental claims, and committing to further improvements. Given all positive scores, the authors' effective responses, and the paper's potential impact on the research community, I recommend accepting the paper if there is sufficient space at the conference.